# A Unified Contrastive Energy-based Model for Understanding the Generative Ability of Adversarial Training

**Yifei Wang**[1]     **Yisen Wang**[2,3*]     **Jiansheng Yang**[1]     **Zhouchen Lin**[2,3,4]

[1] School of Mathematical Sciences, Peking University
[2] Key Lab. of Machine Perception (MoE), School of Artificial Intelligence, Peking University
[3] Institute for Artificial Intelligence, Peking University
[4] Pazhou Lab, Guangzhou, 510330, China

## Abstract

Adversarial Training (AT) is known as an effective approach to enhance the robustness of deep neural networks. Recently researchers notice that robust models with AT have good generative ability and can synthesize realistic images, while the reason behind it is yet under-explored. In this paper, we demystify this phenomenon by developing a unified probabilistic framework, called Contrastive Energy-based Models (CEM). On the one hand, we provide the first probabilistic characterization of AT through a unified understanding of robustness and generative ability. On the other hand, our unified framework can be extended to the unsupervised scenario, which interprets unsupervised contrastive learning as an important sampling of CEM. Based on these, we propose a principled method to develop adversarial learning and sampling methods. Experiments show that the sampling methods derived from our framework improve the sample quality in both supervised and unsupervised learning. Notably, our unsupervised adversarial sampling method achieves an Inception score of 9.61 on CIFAR-10, which is superior to previous energy-based models and comparable to state-of-the-art generative models.

## 1 Introduction

Adversarial Training (AT) is one of the most effective approaches developed so far to improve the robustness of deep neural networks (DNNs) (Madry et al., 2018). AT solves a minimax optimization problem, with the *inner maximization* generating adversarial examples by maximizing the classification loss, and the *outer minimization* finding model parameters by minimizing the loss on adversarial examples generated from the inner maximization (Wang et al., 2019). Recently, researchers have noticed that such robust classifiers obtained by AT are able to extract features that are perceptually aligned with humans (Engstrom et al., 2019). Furthermore, they are able to synthesize realistic images on par with state-of-the-art generative models (Santurkar et al., 2019). Nevertheless, it is still a mystery why AT is able to learn more semantically meaningful features and turn classifiers into generators. Besides, AT needs the labeled data $\{(\mathbf{x}_i, \mathbf{y}_i)\}$ for training while canonical deep generative models do not, *e.g.*, VAE (Kingma & Welling, 2014) and GAN (Goodfellow et al., 2015) only require $\{\mathbf{x}_i\}$. Thus, it is worth exploring if it is possible to train a robust model without labeled data. Several recent works (Jiang et al., 2020; Kim et al., 2020; Ho & Vasconcelos, 2020) have proposed unsupervised AT by adversarially attacking the InfoNCE loss (Oord et al., 2018) (a widely used objective in unsupervised contrastive learning), which indeed improves the robustness of contrastive encoders. However, a depth investigation and understanding for unsupervised AT is still missing.

To address the above issues, in this work, we propose a unified probabilistic framework, Contrastive Energy-based Models (CEM), that provides a principled understanding on the robustness and the generative ability of different training paradigms. Specifically, we make the following contributions:

- **Demystifying adversarial training and sampling.** We firstly propose a probabilistic interpretation for AT, that is, it is inherently a (biased) maximum likelihood training of the

---

*Corresponding author: Yisen Wang (yisen.wang@pku.edu.cn).

corresponding energy-based model, which explains the generative ability of robust models learned by AT. Inspired by this, we propose some novel sampling algorithms with better sample quality than previous methods.

- **A unified probabilistic framework.** Based on the understanding above, we propose Contrastive Energy-based Model (CEM) that incorporates both supervised and unsupervised learning paradigms. Our CEM provides a unified probabilistic understanding of previous standard and adversarial training methods in both supervised and unsupervised learning.

- **Principled unsupervised adversarial training and sampling.** Specifically, under our proposed CEM framework, we establish the equivalence between the importance sampling of CEM and the InfoNCE loss of contrastive learning, which enables us to design principled adversarial sampling for unsupervised learning.

Notably, we show that the sampling methods derived from our framework achieve state-of-the-art sample quality (9.61 Inception score) with unsupervised robust models, which is comparable to both the supervised counterparts and other state-of-the-art generative models.

## 2  RELATED WORK

**Robust generative models.** Researchers recently notice that features extracted by robust classifiers are perceptually aligned with humans, while standard classifiers are not (Engstrom et al., 2019; Kaur et al., 2019; Bai et al., 2021). Santurkar et al. (2019) show that we can also generate images of high quality with robust classifiers by iterative updating from a randomly sampled noise, where the resulting sample quality is comparable to the state-of-the-art generative models like BigGAN (Brock et al., 2018).

**Contrastive learning.** Oord et al. (2018) firstly propose unsupervised contrastive learning by maximizing a tractable lower bound on mutual information (MI), *i.e.,* the negative InfoNCE loss. However, later works find that the lower bounds degrade a lot with a large MI, and the success of these methods cannot be attributed to the properties of MI alone (Poole et al., 2019; Tschannen et al., 2020). Our work provides an alternative understanding of unsupervised contrastive learning as importance sampling of an energy-based model, which also enables us to characterize the limitations of existing methods from a new perspective. In fact, contrastive learning can also be seen as a general learning framework beyond the unsupervised scenarios. For example, SupContrast (Khosla et al., 2020) extends contrastive learning to supervised scenarios. Our work further bridges supervised, unsupervised and adversarial contrastive learning with a unified probabilistic framework.

## 3  CEM: A UNIFIED PROBABILISTIC FRAMEWORK

Inspired by previous work that bridges discriminative models with energy-based models (Grathwohl et al., 2019), in this work, we propose a unified framework, called Contrastive Energy-based Model (CEM), that incorporates both supervised and unsupervised scenarios.

Our proposed CEM is a special Energy-based Model (EBM) that models the joint distribution $p_{\boldsymbol{\theta}}(\mathbf{u}, \mathbf{v})$ over two variables $(\mathbf{u}, \mathbf{v})$ with a similarity function $f_{\boldsymbol{\theta}}(\mathbf{u}, \mathbf{v})$ defined in a contrastive form,

$$p_{\boldsymbol{\theta}}(\mathbf{u}, \mathbf{v}) = \frac{\exp(f_{\boldsymbol{\theta}}(\mathbf{u}, \mathbf{v}))}{Z(\boldsymbol{\theta})}, \tag{1}$$

where $Z(\boldsymbol{\theta}) = \int \exp\left(f_{\boldsymbol{\theta}}(\mathbf{u}, \mathbf{v})\right) d\mathbf{u} d\mathbf{v}$ is the corresponding partition function. In other words, in CEM, a pair of samples $(\mathbf{u}, \mathbf{v})$ has higher probability if they are more alike. In particular, it can be instantiated into the two following variants under different learning scenarios.

**Parametric CEM.** In the supervised scenario, we specify the Parametric CEM (P-CEM) that models the joint distribution $p_{\boldsymbol{\theta}}(\mathbf{x}, y)$ of data $\mathbf{x}$ and label $y$ in the following form,

$$p_{\boldsymbol{\theta}}(\mathbf{x}, y) = \frac{\exp(f_{\boldsymbol{\theta}}(\mathbf{x}, y))}{Z(\boldsymbol{\theta})} = \frac{\exp(g_{\theta}(\mathbf{x})^{\top} \mathbf{w}_y)}{Z(\boldsymbol{\theta})}, \tag{2}$$

where $g_{\boldsymbol{\theta}} : \mathbb{R}^n \to \mathbb{R}^m$ denotes the encoder, $g(\mathbf{x}) \in \mathbb{R}^m$ is the representation of $\mathbf{x}$, and $\mathbf{w}_k \in \mathbb{R}^m$ refers to the parametric cluster center of the $k$-th class. Denote the linear classification weight as

$\mathbf{W} = [\mathbf{w}_1, \cdots, \mathbf{w}_K]$ and the logit vector as $h(\mathbf{x}) = g(\mathbf{x})^\top \mathbf{W}$, we can see the equivalence between P-CEM and JEM (Grathwohl et al., 2019) as

$$f_{\boldsymbol{\theta}}(\mathbf{x}, y) = g_{\boldsymbol{\theta}}(\mathbf{x})^\top \mathbf{w}_y = h_{\boldsymbol{\theta}}(\mathbf{x})[y]. \tag{3}$$

**Non-Parametric CEM.** In the unsupervised scenario, we do not have access to labels, thus we instead model the joint distribution between two natural samples $(\mathbf{x}, \mathbf{x}')$ as

$$p_{\boldsymbol{\theta}}(\mathbf{x}, \mathbf{x}') = \frac{\exp(f_{\boldsymbol{\theta}}(\mathbf{x}, \mathbf{x}'))}{Z(\boldsymbol{\theta})} = \frac{\exp\left(g_{\boldsymbol{\theta}}(\mathbf{x})^\top g_{\boldsymbol{\theta}}(\mathbf{x}')\right)}{Z(\boldsymbol{\theta})}, \tag{4}$$

and the corresponding likelihood gradient of this Non-Parametric CEM (NP-CEM) is

$$\nabla_{\boldsymbol{\theta}} \mathbb{E}_{p_d(\mathbf{x}, \mathbf{x}')} \log p_{\boldsymbol{\theta}}(\mathbf{x}, \mathbf{x}') = \mathbb{E}_{p_d(\mathbf{x}, \mathbf{x}')} \nabla_{\boldsymbol{\theta}} f_{\boldsymbol{\theta}}(\mathbf{x}, \mathbf{x}') - \mathbb{E}_{p_{\boldsymbol{\theta}}(\hat{\mathbf{x}}, \hat{\mathbf{x}}')} \nabla_{\boldsymbol{\theta}} f_{\boldsymbol{\theta}}(\hat{\mathbf{x}}, \hat{\mathbf{x}}'). \tag{5}$$

In contrastive to P-CEM that incorporates parametric cluster centers, the joint distribution of NP-CEM is directly defined based on the feature-level similarity between the two instances $(\mathbf{x}, \mathbf{x}')$. We define the joint data distribution $p_d(\mathbf{x}, \mathbf{x}') = p_d(\mathbf{x}) p_d(\mathbf{x}'|\mathbf{x})$ through re-parameterization,

$$\mathbf{x}' = f_{\boldsymbol{\theta}}(t(\mathbf{x})), \qquad t \overset{u.a.r.}{\sim} \mathcal{T}, \quad \mathbf{x} \sim p_d(\mathbf{x}), \tag{6}$$

where $u.a.r.$ denotes sampling uniformly at random and $\mathcal{T}$ refers to a set of predefined data augmentation operators $\mathcal{T} = \{t : \mathbb{R}^n \to \mathbb{R}^n\}$. For the ease of exposition, we assume the empirical data distribution $p_d(\mathbf{x})$ is uniformly distributed over a finite (but can be exponentially large) set of natural samples $\mathcal{X}$.

# 4 SUPERVISED SCENARIO: REDISCOVERING ADVERSARIAL TRAINING AS MAXIMUM LIKELIHOOD TRAINING

In this section, we investigate why robust models have a good generative ability. The objective of AT is to solve the following *minimax optimization* problem:

$$\min_{\boldsymbol{\theta}} \mathbb{E}_{p_d(\mathbf{x}, y)} \left[ \max_{\|\hat{\mathbf{x}} - \mathbf{x}\|_p \leq \varepsilon} \ell_{CE}(\hat{\mathbf{x}}, y; \boldsymbol{\theta}) \right], \text{ where } \ell_{CE}(\hat{\mathbf{x}}, y; \boldsymbol{\theta}) = -\log p_{\boldsymbol{\theta}}(y|\hat{\mathbf{x}}). \tag{7}$$

The *inner maximization* problem is to find an adversarial example $\hat{\mathbf{x}}$ within the $\ell_p$-norm $\varepsilon$-ball around the natural example $\mathbf{x}$ that maximizes the CE loss. While the *outer minimization* problem is to find model parameters that minimize the loss on the adversarial examples $\hat{\mathbf{x}}$.

## 4.1 MAXIMIZATION PROCESS

For the inner maximization problem, Projected Gradient Descent (PGD) (Madry et al., 2018) is the commonly used method, which generates the adversarial example $\hat{\mathbf{x}}$ by maximizing the CE loss[1] (*i.e.,* minimizing the log conditional probability) starting from $\hat{\mathbf{x}}_0 = \mathbf{x}$:

$$\hat{\mathbf{x}}_{n+1} = \hat{\mathbf{x}}_n + \alpha \nabla_{\hat{\mathbf{x}}_n} \ell(\hat{\mathbf{x}}_n, y; \boldsymbol{\theta}) = \hat{\mathbf{x}}_n - \alpha \nabla_{\hat{\mathbf{x}}_n} \log p_{\boldsymbol{\theta}}(y|\hat{\mathbf{x}}_n)$$

$$= \hat{\mathbf{x}}_n + \alpha \nabla_{\hat{\mathbf{x}}_n} \left[ \log \sum_{k=1}^K \exp(f_{\boldsymbol{\theta}}(\hat{\mathbf{x}}_n, k)) \right] - \alpha \nabla_{\hat{\mathbf{x}}_n} f_{\boldsymbol{\theta}}(\hat{\mathbf{x}}_n, y), \tag{8}$$

while the Langevin dynamics for sampling P-CEM starts from random noise $\hat{\mathbf{x}}_0 = \boldsymbol{\delta}$ and updates with

$$\hat{\mathbf{x}}_{n+1} = \hat{\mathbf{x}}_n + \alpha \nabla_{\hat{\mathbf{x}}} \log p_{\boldsymbol{\theta}}(\hat{\mathbf{x}}_n) + \sqrt{2\alpha} \cdot \boldsymbol{\varepsilon} \tag{9}$$

$$= \hat{\mathbf{x}}_n + \alpha \nabla_{\hat{\mathbf{x}}_n} \left[ \log \sum_{k=1}^K \exp(f_{\boldsymbol{\theta}}(\hat{\mathbf{x}}_n, k)) \right] + \sqrt{2\alpha} \cdot \boldsymbol{\varepsilon}.$$

Eqns. 8 and 9 both have a positive $\mathrm{logsumexp}$ gradient (the second term) to push up the marginal probability $p_{\boldsymbol{\theta}}(\hat{\mathbf{x}})$. As for the third term, PGD starts from a data point $(\mathbf{x}, y)$ such that it requires the repulsive gradient to be away from the original data point and do the exploration in a local

---

[1]Note that we omit the projection operation and the gradient re-normalization steps.

region. Langevin dynamics instead starts from a random noise and an additive noise $\varepsilon$ is injected for exploration.

**Comparing PGD and Langevin.** Following the above analysis, the maximization process in AT can be seen as a (biased) sampling method that draws samples from the corresponding probabilistic model $p_{\boldsymbol{\theta}}(\hat{\mathbf{x}})$. Compared to Langevin dynamics, PGD imposes specific inductive bias for sampling. With the additional repulsive gradient and $\varepsilon$-ball constraint, it explicitly encourages the samples to be misclassified around the original data points. In practice, adversarial training with such adversarial examples is generally more stable than training JEM with Langevin samples, which indicates that PGD attack is a competitive alternative for the negative sampling method for JEM training.

## 4.2 MINIMIZATION PROCESS

To begin with, the gradient of the joint log likelihood for P-CEM can be written as follows:

$$
\begin{aligned}
&\nabla_{\boldsymbol{\theta}} \mathbb{E}_{p_d(\mathbf{x},y)} \log p_{\boldsymbol{\theta}}(\mathbf{x}, y) \\
=&\mathbb{E}_{p_d(\mathbf{x},y)} \nabla_{\boldsymbol{\theta}} f_{\boldsymbol{\theta}}(\mathbf{x}, y) - \mathbb{E}_{p_{\boldsymbol{\theta}}(\hat{\mathbf{x}},\hat{y})} \nabla_{\boldsymbol{\theta}} f_{\boldsymbol{\theta}}(\hat{\mathbf{x}}, \hat{y}) \\
=&\mathbb{E}_{p_d(\mathbf{x},y)} \nabla_{\boldsymbol{\theta}} f_{\boldsymbol{\theta}}(\mathbf{x}, y) - \mathbb{E}_{p_{\boldsymbol{\theta}}(\hat{\mathbf{x}})p_{\boldsymbol{\theta}}(\hat{y}|\hat{\mathbf{x}})} \nabla_{\boldsymbol{\theta}} f_{\boldsymbol{\theta}}(\hat{\mathbf{x}}, \hat{y}),
\end{aligned}
\tag{10}
$$

where $(\mathbf{x}, y) \sim p_d(\mathbf{x}, y)$ denotes the positive data pair, and $(\hat{\mathbf{x}}, \hat{y}) \sim p_{\boldsymbol{\theta}}(\hat{\mathbf{x}}, \hat{y})$ denotes the negative sample pair. As discussed above, the adversarial examples $\hat{\mathbf{x}}$ generated by the maximization process can be regarded as negative samples, and $\hat{y} \sim p_{\boldsymbol{\theta}}(\hat{y}|\hat{\mathbf{x}})$ denotes the predicted label of $\hat{\mathbf{x}}$. To see how the maximum likelihood training of P-CEM is related to the minimization process of AT, we add an interpolated adversarial pair $(\hat{\mathbf{x}}, y)$ into Eq. 10 and decompose it as the consistency gradient and the contrastive gradient:

$$
\begin{aligned}
&\nabla_{\boldsymbol{\theta}} \mathbb{E}_{p_d(\mathbf{x},y)} \log p_{\boldsymbol{\theta}}(\mathbf{x}, y) = \mathbb{E}_{p_d(\mathbf{x},y) \otimes p_{\boldsymbol{\theta}}(\hat{\mathbf{x}},\hat{y})} \left[ \nabla_{\boldsymbol{\theta}} f_{\boldsymbol{\theta}}(\mathbf{x}, y) - \nabla_{\boldsymbol{\theta}} f_{\boldsymbol{\theta}}(\hat{\mathbf{x}}, \hat{y}) \right] \\
&=\mathbb{E}_{p_d(\mathbf{x},y) \otimes p_{\boldsymbol{\theta}}(\hat{\mathbf{x}},\hat{y})} \Big[ \underbrace{\nabla_{\boldsymbol{\theta}} f_{\boldsymbol{\theta}}(\mathbf{x}, y) - \nabla_{\boldsymbol{\theta}} f_{\boldsymbol{\theta}}(\hat{\mathbf{x}}, y)}_{\text{consistency gradient}} + \underbrace{\nabla_{\boldsymbol{\theta}} f_{\boldsymbol{\theta}}(\hat{\mathbf{x}}, y) - \nabla_{\boldsymbol{\theta}} f_{\boldsymbol{\theta}}(\hat{\mathbf{x}}, \hat{y})}_{\text{contrastive gradient}} \Big].
\end{aligned}
\tag{11}
$$

Next, we show that the two parts correspond to two effective mechanisms developed in the adversarial training literature.

**AT loss.** As the two sample pairs in the contrastive gradient share the same input $\hat{\mathbf{x}}$, we can see that the contrastive gradient can be written equivalently as

$$
\begin{aligned}
&\mathbb{E}_{p_d(\mathbf{x},y) \otimes p_{\boldsymbol{\theta}}(\hat{\mathbf{x}},\hat{y})} \left[ \nabla_{\boldsymbol{\theta}} f_{\boldsymbol{\theta}}(\hat{\mathbf{x}}, y) - \nabla_{\boldsymbol{\theta}} f_{\boldsymbol{\theta}}(\hat{\mathbf{x}}, \hat{y}) \right] \\
=&\mathbb{E}_{p_d(\mathbf{x},y) \otimes p_{\boldsymbol{\theta}}(\hat{\mathbf{x}})} \left[ \nabla_{\boldsymbol{\theta}} f_{\boldsymbol{\theta}}(\hat{\mathbf{x}}, y) - \mathbb{E}_{p_{\boldsymbol{\theta}}(\hat{y}|\hat{\mathbf{x}})} \nabla_{\boldsymbol{\theta}} f_{\boldsymbol{\theta}}(\hat{\mathbf{x}}, \hat{y}) \right] \\
=&\mathbb{E}_{p_d(\mathbf{x},y) \otimes p_{\boldsymbol{\theta}}(\hat{\mathbf{x}})} \nabla_{\boldsymbol{\theta}} \log p_{\boldsymbol{\theta}}(y|\hat{\mathbf{x}}),
\end{aligned}
\tag{12}
$$

which is exactly the negative gradient of the robust CE loss (AT loss) in Eq. 7, in other words, gradient ascent with the contrastive gradient is equivalent to gradient descent *w.r.t.* the AT loss.

**Regularization.** As for the consistency gradient, original AT (Madry et al., 2018) simply ignores it. Its variant TRADES (Zhang et al., 2019) instead proposes the KL regularization $\mathrm{KL}(p(\cdot|\hat{\mathbf{x}})\|p(\cdot|\mathbf{x}))$ that regularizes the consistency of the predicted probabilities on all classes, whose optimum implies that $p(\cdot|\hat{\mathbf{x}}) = p(\cdot|\mathbf{x}) \rightarrow f_{\boldsymbol{\theta}}(\mathbf{x}, y) = f_{\boldsymbol{\theta}}(\hat{\mathbf{x}}, y)$.

**Comparing AT and JEM training paradigms.** The above analysis indicates that the minimization objective of AT is closely related to the maximum likelihood training of JEM (Grathwohl et al., 2019). Compared to JEM that decomposes the joint likelihood into an unconditional model $p_{\boldsymbol{\theta}}(\mathbf{x})$ and a discriminative model $p_{\boldsymbol{\theta}}(y|\mathbf{x})$, the decomposition of AT in Eq. 10 instead stabilizes training by introducing an intermediate adversarial pair $(\hat{\mathbf{x}}, y)$ that bridges the positive pair $(\mathbf{x}, y)$ and the negative pair $(\hat{\mathbf{x}}, \hat{y})$. Besides, it can inject the adversarial robustness bias by regularizing the consistency gradient. Together with our analysis on the maximization process, we show that AT is a competitive alternative for training JEM (a generative model) with more stable training behaviors. That explains why robust models with AT are also generative.

## 4.3 PROPOSED SUPERVISED ADVERSARIAL SAMPLING ALGORITHMS

Our interpretation also inspires principled designs of sampling algorithms for robust classifiers.

**Targeted Attack (TA).** Previously, to draw samples from a robust classifier, Santurkar et al. (2019) utilize targeted attack that optimizes an random initialized input $\hat{\mathbf{x}}_0$ towards a specific class $\hat{y}$:

$$\hat{\mathbf{x}}_{n+1} = \hat{\mathbf{x}}_n + \alpha \nabla_{\mathbf{x}_n} \log p_{\boldsymbol{\theta}}(\hat{y}|\hat{\mathbf{x}}_n) = \hat{\mathbf{x}}_n + \alpha \nabla_{\mathbf{x}} f(\hat{\mathbf{x}}_n, \hat{y}) - \alpha \nabla_{\hat{\mathbf{x}}_n} \left[ \log \sum_{k=1}^{K} \exp(f_{\boldsymbol{\theta}}(\hat{\mathbf{x}}_n, k)) \right]. \quad (13)$$

Compared to PGD attack in Eq. 8, while pushing $\hat{\mathbf{x}}$ towards $\hat{y}$, TA has a negative logsumexp gradient that decreases the marginal probability $p_{\boldsymbol{\theta}}(\hat{\mathbf{x}})$. This could explain why TA is less powerful for adversarial attack and is rarely used for adversarial training.

**Conditional Sampling (CS).** To overcome the drawback of targeted attack, a natural idea would be dropping the negative logsumexp gradient. In fact, we can show that this is equivalent to sampling from the conditional distribution:

$$p_{\boldsymbol{\theta}}(\mathbf{x}|\hat{y}) = \frac{\exp(f_{\boldsymbol{\theta}}(\mathbf{x}, \hat{y}))}{Z_{\mathbf{x}|\hat{y}}(\boldsymbol{\theta})}, \;\; Z_{\mathbf{x}|\hat{y}}(\boldsymbol{\theta}) = \int_{\mathbf{x}} \exp(f_{\boldsymbol{\theta}}(\mathbf{x}, \hat{y}))d\mathbf{x},$$

and its Langevin dynamics takes the form:

$$\hat{\mathbf{x}}_{n+1} = \mathbf{x}_n + \alpha \nabla_{\hat{\mathbf{x}}_n} \log p_{\boldsymbol{\theta}}(\hat{\mathbf{x}}_n|\hat{y}) + \sqrt{2\alpha} \cdot \boldsymbol{\varepsilon} = \hat{\mathbf{x}}_n + \alpha \nabla_{\hat{\mathbf{x}}_n} f_{\boldsymbol{\theta}}(\hat{\mathbf{x}}_n, \hat{y}) + \sqrt{2\alpha} \cdot \boldsymbol{\varepsilon}. \quad (14)$$

Samples drawn this way essentially follow an approximated model distribution, $p_{\boldsymbol{\theta}}(\hat{\mathbf{x}}, \hat{y}) \approx p_d(\hat{y})p_{\boldsymbol{\theta}}(\hat{\mathbf{x}}|\hat{y})$. Thus, CS can be seen as a debiased targeted attack algorithm.

**Reinforced Conditional Sampling (RCS).** Inspired by the above analysis, we can design a biased sampling method that deliberately injects a positive logsumexp gradient:

$$\hat{\mathbf{x}}_{n+1} = \hat{\mathbf{x}}_n + \alpha \nabla_{\hat{\mathbf{x}}_n} f_{\boldsymbol{\theta}}(\hat{\mathbf{x}}_n, \hat{y}) + \alpha \nabla_{\hat{\mathbf{x}}_n} \left[ \log \sum_{k=1}^{K} \exp(f_{\boldsymbol{\theta}}(\hat{\mathbf{x}}_n, k)) \right] + \sqrt{2\alpha} \cdot \boldsymbol{\varepsilon}. \quad (15)$$

With our designed bias, RCS will sample towards the target class $\hat{y}$ by maximizing $p_{\boldsymbol{\theta}}(\hat{\mathbf{x}}|\hat{y})$ (with the $f_{\boldsymbol{\theta}}(\hat{\mathbf{x}}_n, \hat{y})$ term), and at the same time improve the marginal probability $p_{\boldsymbol{\theta}}(\hat{\mathbf{x}})$ (with the logsumexp term). As shown in our experiment, RCS indeed obtains improved sample quality.

### 4.4 Discussion on Standard Training

In the above discussion, we have explained why adversarial training is generative from the perspective of CEM. In fact, it can also help characterize why classifiers with Standard Training (ST) are not generative (*i.e.,* poor sample quality). A key insight is that if we replace the model distribution $p_{\boldsymbol{\theta}}(\hat{\mathbf{x}})$ with the data distribution $p_d(\mathbf{x})$ in Eq. 10, we have

$$\nabla_{\boldsymbol{\theta}} \mathbb{E}_{p_d(\mathbf{x},y)} \log p_{\boldsymbol{\theta}}(\mathbf{x}, y) = \mathbb{E}_{p_d(\mathbf{x},y)} \nabla_{\boldsymbol{\theta}} f_{\boldsymbol{\theta}}(\mathbf{x}, y) - \mathbb{E}_{p_{\boldsymbol{\theta}}(\hat{\mathbf{x}})p_{\boldsymbol{\theta}}(\hat{y}|\hat{\mathbf{x}})} \nabla_{\boldsymbol{\theta}} f_{\boldsymbol{\theta}}(\hat{\mathbf{x}}, \hat{y})$$

$$\approx \mathbb{E}_{p_d(\mathbf{x},y)} \nabla_{\boldsymbol{\theta}} f_{\boldsymbol{\theta}}(\mathbf{x}, y) - \mathbb{E}_{p_d(\mathbf{x})p_{\boldsymbol{\theta}}(\hat{y}|\mathbf{x})} \nabla_{\boldsymbol{\theta}} f_{\boldsymbol{\theta}}(\mathbf{x}, \hat{y}) = \nabla_{\boldsymbol{\theta}} \mathbb{E}_{p_d(\mathbf{x},y)} \log p_{\boldsymbol{\theta}}(y|\mathbf{x}), \quad (16)$$

which is the negative gradient of the CE loss in Eq. 7. Thus, ST is equivalent to training CEM by simply replacing model-based negative samples $\hat{\mathbf{x}} \sim p_{\boldsymbol{\theta}}(\mathbf{x})$ with data samples $\mathbf{x} \sim p_d(\mathbf{x})$. This approximation makes ST computationally efficient with good accuracy on natural data, but significantly limits its robustness on adversarial examples (as model-based negative samples). Similarly, because ST ignores exploring negative samples while training, standard classifiers also fail to generate realistic samples.

## 5 Extension of Adversarial Training to Unsupervised Scenario

In this section, we show that with our unified framework, we can naturally extend the interpretation developed for supervised adversarial training to the unsupervised scenario.

### 5.1 Understanding Unsupervised Standard Training through CEM

**InfoNCE.** Recently, the following InfoNCE loss is widely adopted for unsupervised contrastive learning of data representations (Oord et al., 2018; Chen et al., 2020; He et al., 2020),

$$\ell_{NCE}(\mathbf{x}, \mathbf{x}', \{\hat{\mathbf{x}}_j\}_{j=1}^{K}; \boldsymbol{\theta}) = -\log \frac{\exp(f_{\boldsymbol{\theta}}(\mathbf{x}, \mathbf{x}'))}{\sum_{i=j}^{K} \exp(f_{\boldsymbol{\theta}}(\mathbf{x}, \hat{\mathbf{x}}_j))}, \quad (17)$$

where $f_{\boldsymbol{\theta}}(\mathbf{x}, \hat{\mathbf{x}}) = g_{\boldsymbol{\theta}}(\mathbf{x})^{\top} g_{\boldsymbol{\theta}}(\hat{\mathbf{x}})$ calculates the similarity between the representations of the two data samples, $\mathbf{x}, \mathbf{x}'$ are generated by two random augmentations (drawn from $\mathcal{T}$) of the same data example, and $\{\hat{\mathbf{x}}_j\}_{j=1}^{K}$ denotes $K$ independently drawn negative samples. In practice, one of the $K$ negative samples is chosen to be the positive sample $\mathbf{x}'$. Therefore, InfoNCE can be seen as an instance-wise $K$-class cross entropy loss for non-parametric classification.

Perhaps surprisingly, we show that the InfoNCE loss is equivalent to the importance sampling estimate of our NP-CEM (Eq. 4) by approximating the negative samples from $p_{\boldsymbol{\theta}}(\mathbf{x})$ with data samples from $p_d(\mathbf{x})$, as what we have done in standard supervised training (Section 4.4):

$$\mathbb{E}_{p_d(\mathbf{x},\mathbf{x}')}\nabla_{\boldsymbol{\theta}} f_{\boldsymbol{\theta}}(\mathbf{x}, \mathbf{x}') - \mathbb{E}_{p_{\boldsymbol{\theta}}(\hat{\mathbf{x}},\hat{\mathbf{x}}')}\nabla_{\boldsymbol{\theta}} f_{\boldsymbol{\theta}}(\hat{\mathbf{x}}, \hat{\mathbf{x}}')$$

$$=\mathbb{E}_{p_d(\mathbf{x},\mathbf{x}')}\nabla_{\boldsymbol{\theta}} f_{\boldsymbol{\theta}}(\mathbf{x}, \mathbf{x}') - \mathbb{E}_{p_{\boldsymbol{\theta}}(\hat{\mathbf{x}})p_d(\hat{\mathbf{x}}')}\frac{\exp(f_{\boldsymbol{\theta}}(\hat{\mathbf{x}}, \hat{\mathbf{x}}'))}{\mathbb{E}_{p_d(\tilde{\mathbf{x}})}\exp(f_{\boldsymbol{\theta}}(\hat{\mathbf{x}}, \tilde{\mathbf{x}}))}\nabla_{\boldsymbol{\theta}} f_{\boldsymbol{\theta}}(\hat{\mathbf{x}}, \hat{\mathbf{x}}')$$

$$\approx\mathbb{E}_{p_d(\mathbf{x},\mathbf{x}')}\nabla_{\boldsymbol{\theta}} f_{\boldsymbol{\theta}}(\mathbf{x}, \mathbf{x}') - \mathbb{E}_{p_d(\hat{\mathbf{x}})p_d(\hat{\mathbf{x}}')}\frac{\exp(f_{\boldsymbol{\theta}}(\hat{\mathbf{x}}, \hat{\mathbf{x}}'))}{\mathbb{E}_{p_d(\tilde{\mathbf{x}})}\exp(f_{\boldsymbol{\theta}}(\hat{\mathbf{x}}, \tilde{\mathbf{x}}))}\nabla_{\boldsymbol{\theta}} f_{\boldsymbol{\theta}}(\hat{\mathbf{x}}, \hat{\mathbf{x}}') \qquad (18)$$

$$=\mathbb{E}_{p_d(\mathbf{x},\mathbf{x}')}\nabla_{\boldsymbol{\theta}} \log \frac{\exp(f_{\boldsymbol{\theta}}(\mathbf{x}, \mathbf{x}'))}{\mathbb{E}_{p_d(\hat{\mathbf{x}}')}\exp(f_{\boldsymbol{\theta}}(\mathbf{x}, \hat{\mathbf{x}}'))} \approx \frac{1}{N}\sum_{i=1}^{N}\nabla_{\boldsymbol{\theta}} \log \frac{\exp(f_{\boldsymbol{\theta}}(\mathbf{x}_i, \mathbf{x}_i'))}{\sum_{k=1}^{K}\exp(f_{\boldsymbol{\theta}}(\mathbf{x}_i, \hat{\mathbf{x}}_{ik}'))},$$

which is exactly the negative gradient of the InfoNCE loss. In the above analysis, for an empirical estimate, we draw $N$ positive pairs $(\mathbf{x}_i, \mathbf{x}_i') \sim p_d(\mathbf{x}, \mathbf{x}')$, and for each anchor $\mathbf{x}_i$, we further draw $K$ negative samples $\{\hat{\mathbf{x}}_{ik}'\}$ independently from $p_d(\hat{\mathbf{x}}')$.

**Remark.** As $p_{\boldsymbol{\theta}}(\hat{\mathbf{x}}, \hat{\mathbf{x}}') = p_{\boldsymbol{\theta}}(\hat{\mathbf{x}})p_{\boldsymbol{\theta}}(\hat{\mathbf{x}}'|\hat{\mathbf{x}})$, the negative phase of NP-CEM is supposed to sample $\hat{\mathbf{x}}'$ from $p_{\boldsymbol{\theta}}(\hat{\mathbf{x}}'|\hat{\mathbf{x}})$, where samples semantically close to the anchor sample $\hat{\mathbf{x}}$, a.k.a. hard negative samples, should have high probabilities. However, InfoNCE adopts a *non-informative* uniform proposal $p_d(\hat{\mathbf{x}}')$ for importance sampling, which is very sample inefficient because most samples are useless (Kalantidis et al., 2020). This observation motivates us to design more efficient sampling scheme for contrastive learning by mining hard negatives. For example, Robinson et al. (2021) directly replace the plain proposal with $\tilde{p}_{\boldsymbol{\theta}}(\hat{\mathbf{x}}|\hat{\mathbf{x}}') = \exp(\beta f_{\boldsymbol{\theta}}(\hat{\mathbf{x}}, \hat{\mathbf{x}}'))/Z_{\beta}(\boldsymbol{\theta})$ while keeping the reweighing term. From the perspective of CEM, the temperature $\beta$ introduces bias that should be treated carefully. In all, CEM provides a principled framework to develop efficient contrastive learning algorithms.

## 5.2 PROPOSED UNSUPERVISED ADVERSARIAL TRAINING

AT is initially designed for supervised learning, where adversarial examples can be clearly defined by misclassification. However, it remain unclear what is the right way to do Unsupervised Adversarial Training (UAT) without access to *any* labels. Previous works (Jiang et al., 2020; Ho & Vasconcelos, 2020; Kim et al., 2020) have carried out UAT with the adversarial InfoNCE loss, which works well but lacks theoretical justification. Our unified CEM framework offers a principled way to generalize adversarial training from supervised to unsupervised scenarios.

**Maximization Process.** Sampling from $p_{\boldsymbol{\theta}}(\mathbf{x})$ can be more difficult than that in supervised scenarios because it does not admit a closed form for variable $\mathbf{x}'$. Thus, we perform Langevin dynamics with $K$ negative samples $\{\hat{\mathbf{x}}_k'\}$ drawn from $p_d(\hat{\mathbf{x}}')$,

$$\hat{\mathbf{x}}_{n+1} = \hat{\mathbf{x}}_n + \alpha\nabla_{\hat{\mathbf{x}}_n} \log p_{\boldsymbol{\theta}}(\hat{\mathbf{x}}_n) + \sqrt{2\alpha}\cdot\boldsymbol{\varepsilon} \qquad (19)$$

$$\approx \hat{\mathbf{x}}_n + \alpha\nabla_{\hat{\mathbf{x}}_n}\left[\log\frac{1}{K}\sum_{k=1}^{K}p_{\boldsymbol{\theta}}(\hat{\mathbf{x}}_n, \hat{\mathbf{x}}_k')\right] + \sqrt{2\alpha}\cdot\boldsymbol{\varepsilon}$$

$$= \hat{\mathbf{x}}_n + \alpha\nabla_{\hat{\mathbf{x}}_n}\left[\log\sum_{k=1}^{K}\exp(f_{\boldsymbol{\theta}}(\hat{\mathbf{x}}_n, \hat{\mathbf{x}}_k'))\right] + \sqrt{2\alpha}\cdot\boldsymbol{\varepsilon}.$$

While the PGD attack of the InfoNCE loss (Eq. 31),

$$\hat{\mathbf{x}}_{n+1} = \hat{\mathbf{x}}_n + \alpha\nabla_{\hat{\mathbf{x}}_n} \log\frac{\exp(f_{\boldsymbol{\theta}}(\hat{\mathbf{x}}_n, \mathbf{x}'))}{\sum_{k=1}^{K}\exp(f_{\boldsymbol{\theta}}(\hat{\mathbf{x}}_n, \hat{\mathbf{x}}_k'))} \qquad (20)$$

$$= \hat{\mathbf{x}}_n + \alpha\nabla_{\hat{\mathbf{x}}_n}\left[\log\sum_{k=1}^{K}\exp(f_{\boldsymbol{\theta}}(\hat{\mathbf{x}}_n, \hat{\mathbf{x}}_k'))\right] - \alpha\nabla_{\boldsymbol{\theta}} f_{\boldsymbol{\theta}}(\hat{\mathbf{x}}_n, \mathbf{x}'),$$

resembles the Langevin dynamics as they both share the positive $\mathrm{logsumexp}$ gradient that pushes up $p_{\boldsymbol{\theta}}(\hat{\mathbf{x}})$, and differs by a repulse negative gradient $-f_{\boldsymbol{\theta}}(\hat{\mathbf{x}}, \mathbf{x}')$ away from the anchor $\mathbf{x}'$, which is a direct analogy of the PGD attack in supervised learning (Section 4.1). Therefore, we believe that the PGD attack of InfoNCE is a proper way to craft adversarial examples by sampling from $p_{\boldsymbol{\theta}}(\mathbf{x})$.

**Minimization Process.** Following the same routine in Section 4.2, with the adversarial example $\hat{\mathbf{x}} \sim p_{\boldsymbol{\theta}}(\hat{\mathbf{x}})$, we can insert an interpolated adversarial pair $(\hat{\mathbf{x}}, \mathbf{x}')$ and decompose the gradient of NP-CEM into the consistency gradient and the contrastive gradient,

$$
\nabla_{\boldsymbol{\theta}} \mathbb{E}_{p_d(\mathbf{x}, \mathbf{x}')} \log p_{\boldsymbol{\theta}}(\mathbf{x}, \mathbf{x}') = \mathbb{E}_{p_d(\mathbf{x}, \mathbf{x}') \otimes p_{\boldsymbol{\theta}}(\hat{\mathbf{x}}, \hat{\mathbf{x}}')} [\nabla_{\boldsymbol{\theta}} f_{\boldsymbol{\theta}}(\mathbf{x}, \mathbf{x}') - \nabla_{\boldsymbol{\theta}} f_{\boldsymbol{\theta}}(\hat{\mathbf{x}}, \mathbf{x}')]
$$

$$
= \mathbb{E}_{p_d(\mathbf{x}, \mathbf{x}') \otimes p_{\boldsymbol{\theta}}(\hat{\mathbf{x}}, \hat{\mathbf{x}}')} \Big[ \underbrace{\nabla_{\boldsymbol{\theta}} f_{\boldsymbol{\theta}}(\mathbf{x}, \mathbf{x}') - \nabla_{\boldsymbol{\theta}} f_{\boldsymbol{\theta}}(\hat{\mathbf{x}}, \mathbf{x}')}_{\text{consistency gradient}} + \underbrace{\nabla_{\boldsymbol{\theta}} f_{\boldsymbol{\theta}}(\hat{\mathbf{x}}, \mathbf{x}') - \nabla_{\boldsymbol{\theta}} f_{\boldsymbol{\theta}}(\hat{\mathbf{x}}, \hat{\mathbf{x}}')}_{\text{contrastive gradient}} \Big]. \quad (21)
$$

In this way, we can directly develop the unsupervised analogy of AT loss and regularization (Sec. 4.2). Following Eq. 30, it is easy to see that the contrastive gradient is equivalent to the gradient of the Adversarial InfoNCE loss utilized in previous work (Jiang et al., 2020; Ho & Vasconcelos, 2020; Kim et al., 2020) with adversarial example $\hat{\mathbf{x}}$,

$$
\mathbb{E}_{p_d(\mathbf{x}, \mathbf{x}') \otimes p_{\boldsymbol{\theta}}(\hat{\mathbf{x}}, \hat{\mathbf{x}}')} [\nabla_{\boldsymbol{\theta}} f_{\boldsymbol{\theta}}(\hat{\mathbf{x}}, \mathbf{x}') - \nabla_{\boldsymbol{\theta}} f_{\boldsymbol{\theta}}(\hat{\mathbf{x}}, \hat{\mathbf{x}}')]
$$

$$
= \mathbb{E}_{p_d(\mathbf{x}, \mathbf{x}') \otimes p_{\boldsymbol{\theta}}(\hat{\mathbf{x}})} \left[ \nabla_{\boldsymbol{\theta}} f_{\boldsymbol{\theta}}(\hat{\mathbf{x}}, \mathbf{x}') - \mathbb{E}_{p_{\boldsymbol{\theta}}(\hat{\mathbf{x}}') p_{\boldsymbol{\theta}}(\hat{\mathbf{x}})} \frac{p_{\boldsymbol{\theta}}(\hat{\mathbf{x}}|\hat{\mathbf{x}}')}{p_{\boldsymbol{\theta}}(\hat{\mathbf{x}})} \nabla_{\boldsymbol{\theta}} f_{\boldsymbol{\theta}}(\hat{\mathbf{x}}, \hat{\mathbf{x}}') \right]
$$

$$
= \mathbb{E}_{p_d(\mathbf{x}, \mathbf{x}') \otimes p_{\boldsymbol{\theta}}(\hat{\mathbf{x}})} \left[ \nabla_{\boldsymbol{\theta}} f_{\boldsymbol{\theta}}(\hat{\mathbf{x}}, \mathbf{x}') - \mathbb{E}_{p_{\boldsymbol{\theta}}(\hat{\mathbf{x}}')} \frac{p_{\boldsymbol{\theta}}(\hat{\mathbf{x}}|\hat{\mathbf{x}}')}{p_{\boldsymbol{\theta}}(\hat{\mathbf{x}})} \nabla_{\boldsymbol{\theta}} f_{\boldsymbol{\theta}}(\hat{\mathbf{x}}, \hat{\mathbf{x}}') \right]
$$

$$
\approx \frac{1}{N} \sum_{i=1}^{N} \nabla_{\boldsymbol{\theta}} \log \frac{\exp(f_{\boldsymbol{\theta}}(\hat{\mathbf{x}}_i, \mathbf{x}'_i))}{\sum_{k=1}^{K} \exp(f_{\boldsymbol{\theta}}(\hat{\mathbf{x}}_i, \hat{\mathbf{x}}'_{ik}))}, \quad (22)
$$

where $\{\hat{\mathbf{x}}'_{ik}\}$ denotes the adversarial negative samples from $p_{\boldsymbol{\theta}}(\hat{\mathbf{x}}')$.

### 5.3 PROPOSED UNSUPERVISED ADVERSARIAL SAMPLING

In supervised learning, a natural method to draw a sample $\hat{\mathbf{x}}$ from a robust classifier is to maximize its conditional probability *w.r.t.* a given class $\hat{y} \sim p_d(\hat{y})$, *i.e.,* $\max_{\hat{\mathbf{x}}} p(\hat{y}|\hat{\mathbf{x}})$, by targeted attack (Santurkar et al., 2019). However, in the unsupervised scenarios, we do not have access to labels, and this approach is not applicable anymore. Meanwhile, Langevin dynamics is also not directly applicable (Eq. 19) because it requires access to real data samples.

**MaxEnt.** Nevertheless, we still find an effective algorithm for drawing samples from an unsupervised robust model. We first initialize a batch of $N$ samples $\mathcal{B} = \{\mathbf{x}_i\}_{i=1}^{N}$ from a prior distribution $p_0(\mathbf{x})$ (*e.g.,* Gaussian). Next, we update the batch jointly by maximizing the empirical estimate of entropy, where we simply take the generated samples at $\mathcal{B} = \{\mathbf{x}_i\}_{i=1}^{N}$ as samples from $p_{\boldsymbol{\theta}}(\mathbf{x})$

$$
\mathcal{H}(p_{\boldsymbol{\theta}}) = -\mathbb{E}_{p_{\boldsymbol{\theta}}(\mathbf{x})} \log p_{\boldsymbol{\theta}}(\mathbf{x}) \approx -\frac{1}{N} \sum_{i=1}^{N} p_{\boldsymbol{\theta}}(\mathbf{x}_i) \approx -\frac{1}{N} \sum_{i=1}^{N} \log \frac{1}{N} \sum_{j=1}^{N} \exp(f_{\boldsymbol{\theta}}(\mathbf{x}_i, \mathbf{x}_j)) + \log Z(\boldsymbol{\theta}). \quad (23)
$$

Specifically, we update each sample $\mathbf{x}_i$ by maximizing the empirical entropy (named **MaxEnt**)

$$
\mathbf{x}'_i = \mathbf{x}_i + \alpha \nabla_{\mathbf{x}_i} \mathcal{H}(p_{\boldsymbol{\theta}}) + \sqrt{2\alpha} \cdot \boldsymbol{\varepsilon} \approx \mathbf{x}_i - \alpha \nabla_{\mathbf{x}_i} \sum_{i=1}^{N} \log \sum_{j=1}^{N} \exp(f_{\boldsymbol{\theta}}(\mathbf{x}_i, \mathbf{x}_j)) + \sqrt{2\alpha} \cdot \boldsymbol{\varepsilon}. \quad (24)
$$

As a result, the generated samples $\{\mathbf{x}_i\}_{i=1}^{N}$ are encouraged to distribute uniformly in the feature space with maximum entropy, and thus cover the overall model distribution with diverse semantics.

## 6 EXPERIMENTS

In this section, we evaluate the adversarial sampling methods derived from our CEM framework, showing that they can bring significant improvement to the sample quality of previous work. Besides, in Appendix A, we also conduct a range of experiments on adversarial robustness to verify our probabilistic understandings of AT. We show adversarial training objectives derived from our

Table 1: Inception Scores (IS) and Fréchet Inception Distance (FID) of different generative models. Results marked with ⋆ are taken from Shmelkov et al. (2018).

| Method | IS ($\uparrow$) | FID ($\downarrow$) |
|---|---|---|
| **Auto-regressive** | | |
| PixelCNN++⋆ (Salimans et al., 2017) | 5.36 | 119.5 |
| **GAN-based** | | |
| DCGAN⋆ (Radford et al., 2016) | 6.69 | 35.6 |
| WGAN-GP (Gulrajani et al., 2017) | 7.86 | 36.4 |
| PresGAN (Dieng et al., 2019) | - | 52.2 |
| StyleGAN2-ADA (Karras et al., 2020) | **10.02** | - |
| **Score-based** | | |
| NCSN (Song & Ermon, 2019) | 8.87 | 25.32 |
| DDPM (Ho et al., 2020) | 9.46 | 3.17 |
| NCSN++ (Song et al., 2020) | 9.89 | **2.20** |
| **EBM-based** | | |
| JEM (Grathwohl et al., 2019) | 8.76 | 38.4 |
| DRL (Gao et al., 2021) | 8.58 | 9.60 |
| **AT-based** | | |
| TA (Santurkar et al., 2019) (w/ ResNet50) | 7.5 | - |
| **Supervised CEM** (w/ ResNet50) | **9.80** | 55.91 |
| **Unsupervised CEM** (w/ ResNet18) (ours) | 8.68 | **36.4** |
| **Unsupervised CEM** (w/ ResNet50) (ours) | 9.61 | 40.25 |

CEM can indeed significantly improve the performance of AT in both supervised and unsupervised scenarios, which helps justify our interpretations and our framework.

**Models.** For supervised robust models, we adopt the same pretrained ResNet50 checkpoint[2] on CIFAR-10 as Santurkar et al. (2019) for a fair comparison. The model is adversarially trained with $\ell_2$-norm PGD attack with random start, maximal perturbation norm 0.5, step size 0.1 and 7 steps. As for the unsupervised case, we are the first to consider sampling from unsupervised robust models. We train ResNet18 and ResNet50 (He et al., 2016) following the setup of an existing unsupervised adversarial training method ACL (Jiang et al., 2020). The training attack is kept the same as that of the supervised case for a fair comparison. More details are provided in Appendix.

**Sampling protocol.** In practice, our adversarial sampling methods take the following general form as a mixture of the PGD and Langevin dynamics,

$$\mathbf{x}_{n+1} = \Pi_{\|\mathbf{x}_n - \mathbf{x}_0\|_2 \leq \beta} \left[ \mathbf{x}_n + \alpha \mathbf{g}_k + \eta \boldsymbol{\varepsilon}_k \right], \mathbf{x}_0 = \boldsymbol{\delta}, \boldsymbol{\varepsilon}_k \sim \mathcal{N}(\mathbf{x}'ero, \mathbf{1}), k = 0, 1, \ldots, K,$$

where $\mathbf{g}_k$ is the update gradient, $\boldsymbol{\varepsilon}_k$ is the diffusion noise, $\Pi_{\mathcal{S}}$ is the projector operator, and $\boldsymbol{\delta}$ is the (conditional) initial seeds drawn from the multivariate normal distribution whose mean and covariance are calculated from the CIFAR-10 test set following Santurkar et al. (2019). We evaluate the sample quality quantitatively with Inception Score (IS) (Salimans et al., 2016) and Fréchet Inception Distance (FID) (Heusel et al., 2017). More details can be found in in Appendix C.1.

**Comparison with other generative models**. In Table 1, we compare the sample quality of adversarial sampling methods with different kinds of generative models. We analyze the results in terms of the following aspects:

- Our adversarial sampling methods outperform many deep generative models like Pixel-CNN++, WGAN-GP and PresGAN, and obtain state-of-the-art Inception scores on par with StyleGAN2 (Karras et al., 2020).
- Comparing our AT-based methods with previous methods for training EBMs (Grathwohl et al., 2019; Gao et al., 2021), we see that it obtains state-of-the-art Inception scores among

---

[2]We download the checkpoint from the repository `https://github.com/MadryLab/robustness_applications`.

the EBM-based methods. Remarkably, our unsupervised CEM with ResNet18 obtains both better IS and FID scores than the original JEM, which adopts WideResNet-28-10 (Zagoruyko & Komodakis, 2016) with even more parameters.

- Compared with previous AT-based method (Santurkar et al., 2019), our CEM-based methods also improve IS by a large margin (even with the unsupervised ResNet18). Remarkably, the supervised and unsupervised methods obtain similar sample quality, and the supervised methods are better (higher) at IS while the unsupervised methods are better (lower) at FID.

- We obtain similar Inception scores as state-of-the-art score-based models like NCSN++, while fail to match their FID scores. Nevertheless, a significant drawback of these methods is that they typically require more than 1,000 steps to draw a sample, while ours only require less than 50 steps.

Table 2: Inception Scores (IS) and Fréchet Inception Distance (FID) of different sampling methods for adversarially robust models. Cond: conditional. Uncond: unconditional.

| Training | Sampling | Method | IS (↑) | FID (↓) |
|---|---|---|---|---|
| Supervised | Cond | TA | 9.26 | 56.72 |
| | | Langevin | 9.65 | 63.34 |
| | | CS | 9.77 | 56.26 |
| | | RCS | **9.80** | **55.91** |
| Unsupervised (w/ ResNet18) | Uncond | PGD | 5.35 | 74.27 |
| | | MaxEnt | **8.24** | **41.80** |
| | Cond | PGD | 5.85 | 68.54 |
| | | MaxEnt | **8.68** | **36.44** |
| Unsupervised (w/ ResNet50) | Uncond | PGD | 5.24 | 141.54 |
| | | MaxEnt | **9.57** | **44.86** |
| | Cond | PGD | 5.37 | 137.68 |
| | | MaxEnt | **9.61** | **40.25** |

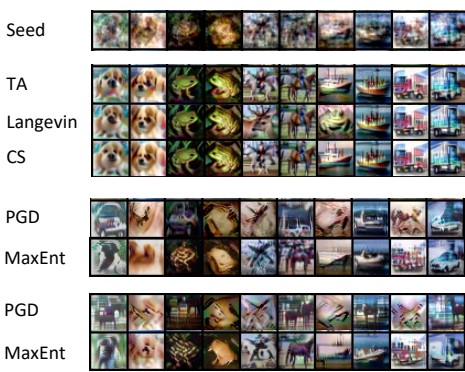

Seed  TA  Langevin  CS  PGD  MaxEnt  PGD  MaxEnt

Figure 1: Four groups of random samples (top to bottom): initial, supervised (ResNet50), unsupervised (ResNet18), unsupervised (ResNet50).

**Comparison among adversarial sampling methods.** In Table 2, we further compare the sample quality of different adversarial sampling methods discussed in Sections 4.3 & 5.3. For supervised models, we can see that indeed TA obtains the lowest IS, while CS can significantly refine the sample quality, and RCS can further improve the sample quality by the injected bias. For unsupervised models, we can see that MaxEnt outperforms PGD consistently by a large margin. In particular, conditional sampling initialized with class-wise noise can improve a little on the sample quality compared to unconditional sampling. The average visual sample quality in Figure 1 is roughly consistent with the quantitative results.

**The mismatch between IS and FID.** A notable issue of adversarial sampling methods is the mismatch between the IS and FID scores. For example, in Table 1, DDPM and our unsupervised CEM (w/ ResNet50) have similar Inception scores, but the FID of DDPM (2.20) is significantly smaller than ours (40.25), a phenomenon also widely observed in previous methods (Santurkar et al., 2019; Grathwohl et al., 2019; Song & Ermon, 2019). Through a visual inspection of the samples in Figure 1, we can see that the samples have realistic global structure, but as for the local structure, we can find some common artifacts, which could be the reason why it has a relatively large distance (FID) to the real data.

## 7 CONCLUSION

In this paper, we proposed a unified probabilistic framework, named Contrastive Energy-based Model (CEM), which not only explains the generative ability of adversarial training, but also provides a unified perspective of adversarial training and sampling in both supervised and unsupervised paradigms. Extensive experiments show that adversarial sampling methods derived from our framework indeed demonstrate better sample quality than state-of-the-art methods.

ACKNOWLEDGEMENT

Yisen Wang is partially supported by the National Natural Science Foundation of China under Grant 62006153, Project 2020BD006 supported by PKU-Baidu Fund, and Open Research Projects of Zhejiang Lab (No. 2022RC0AB05). Jiansheng Yang is supported by the National Science Foundation of China under Grant No. 11961141007. Zhouchen Lin is supported by the NSF China (No. 61731018), NSFC Tianyuan Fund for Mathematics (No. 12026606), Project 2020BD006 supported by PKU-Baidu Fund, and Qualcomm.

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

## A    Evaluating Adversarial Robustness

In Section 6, we have shown that the new adversarial sampling algorithms derived from our CEM framework indeed obtain improved sample quality, which helps justifies our interpretation of AT from a generative aspect. In this section, we further take a complementary way to verify our interpretation by studying its discriminative part. In particular, we develop new variants of AT regularization and verify their effectiveness on improving the adversarial robustness of AT. As CEM has both supervised and unsupervised variants, we develop AT variants for each scenario respectively while following the same spirit.

### A.1    Supervised Adversarial Training

#### A.1.1    Proposed Method

In Section 4.2, we have mentioned that for the consistency gradient, original AT (Madry et al., 2018) simply ignores it. Denoting $\mathbf{x}$ and $\hat{\mathbf{x}}$ as the natural and adversarial inputs, the state-of-the-art AT variant, TRADES (Zhang et al., 2019), instead adopts the KL regularization $\mathrm{KL}(p(\cdot|\hat{\mathbf{x}})\|p(\cdot|\mathbf{x}))$ that explicitly regularizes the consistency of the predicted probabilities on all classes, whose optimum implies that $p(\cdot|\hat{\mathbf{x}}) = p(\cdot|\mathbf{x}) \to f_{\boldsymbol{\theta}}(\mathbf{x}, y) = f_{\boldsymbol{\theta}}(\hat{\mathbf{x}}, y)$.

Inspired by this discussion, alternatively, we can directly regularize the consistency gradient to zero. We achieve this with the following Consistency Regularization (CR) with least square loss:

$$\mathcal{L}_{CR}(\boldsymbol{\theta}) = \mathbb{E}_{p_d(\mathbf{x}, y) \otimes p_{\boldsymbol{\theta}}(\mathbf{x})} \left( f_{\boldsymbol{\theta}}(\mathbf{x}, y) - f_{\boldsymbol{\theta}}(\hat{\mathbf{x}}, y) \right)^2. \tag{25}$$

We note the our proposed AT+CR differs to ALP (Kannan et al., 2018) as we only minimizes the gap between the logits of the label class $f_{\boldsymbol{\theta}}(\mathbf{x}, y)$. ALP was shown to be ineffective for improving AT, while in the experiments below, we show that our AT+CR objective indeed achieves comparable (even slightly better) results as TRADES.

#### A.1.2    Empirical Evaluation

**Experimental setup.** Following the conventions, we compare different AT methods with Preactivated ResNet18 (He et al., 2016) and WideResNet34 (Zagoruyko & Komodakis, 2016) on CIFAR-10. The maximum perturbation is bounded by $\varepsilon = 8/255$ under $\ell_\infty$ norm. The training attack is PGD$^{10}$ (Madry et al., 2018) with random start and step size $\varepsilon/4$. The test attack is PGD$^{20}$ with random start and step size $\varepsilon/4$. We evaluate both the final epoch and the early stopped epoch with the best robust accuracy.

Table 3: Robustness (accuracy (%) on adversarial attacks) of supervised adversarial training methods on CIFAR-10. R18: ResNet18. W34: WideResNet34.

| Model | Training | Natural Acc (%) | Adversarial Acc (%) |
|---|---|---|---|
| ResNet18 | AT (Madry et al., 2018) | **83.7** | 52.2 |
| | TRADES (Zhang et al., 2019) | 82.5 | 54.3 |
| | **AT+CR** (ours) | 81.5 | **55.2** |
| WideResNet34 | AT (Madry et al., 2018) | **86.8** | 53.6 |
| | TRADES (Zhang et al., 2019) | 83.4 | **57.0** |
| | **AT+CR** (ours) | 86.6 | **57.0** |

**Result analysis.** From Table 3, we can see that the AT+CR objective derived from our framework indeed enjoy much better robustness than vanilla AT. Compared to the state-of-the-art AT variant TRADES, we can see that AT+CR is comparable, and sometimes slightly better at robustness. When the two have similar robustness (for WideResNet34), AT+CR obtains better natural accuracy than TRADES (86.6 v.s. 83.4). These results empirically justify that our interpretation of AT and TRADES from a probabilistic perspective.

### A.2    Unsupervised Adversarial Training

Similarly, we can develop the same regularization technique for unsupervised adversarial training through a unified perspective of supervised and unsupervised AT offered by our CEM.

### A.2.1 PROPOSED METHOD

In Section 5.2, we have developed a principled unsupervised adversarial training routine by an analogy with the supervised AT (Section 4.2). Besides, we can also consider an unsupervised version of the consistency regularization above, namely, the Unsupervised Consistency Regularization (UCR),

$$\mathcal{L}_{UCR}(\boldsymbol{\theta}) = \mathbb{E}_{p_d(\mathbf{x},\mathbf{x}') \otimes p_{\boldsymbol{\theta}}(\hat{\mathbf{x}})} \|f_{\boldsymbol{\theta}}(\mathbf{x},\mathbf{x}') - f_{\boldsymbol{\theta}}(\hat{\mathbf{x}},\mathbf{x}')\|^2, \tag{26}$$

which encourages the consistency between the similarity between the natural pair $(\mathbf{x}, \mathbf{x}')$ and the adversarial pair $(\hat{\mathbf{x}}, \mathbf{x}')$.

### A.2.2 EMPIRICAL EVALUATION

**Experimental setup.** Among the many variants of contrastive learning, we adopt SimCLR as our baseline method and take the recently proposed ACL (Jiang et al., 2020) as our Unsupervised Adversarial Training (UAT) following the same default setup as in Section C.1. The training attack configuration is kept the same as the supervised case, while after training, we freeze the encoder and fine-tune a linear classification layer (standard training) on top with labeled data for evaluating the learned features. In particular, we evaluate the composed model on the test data with two different attack methods, FGSM (Goodfellow et al., 2015) (one-step attack) and PGD[20] (multi-step attack), both with $\varepsilon = 8/255$, and report their natural and adversarial accuracy, respectively.

Table 4: Robustness (accuracy (%) on adversarial attacks) of unsupervised contrastive learning methods on CIFAR-10 with ResNet-18 backbone and two different attack methods: FGSM (Goodfellow et al., 2015) and PGD (Madry et al., 2018).

| Training | Natural Acc (%) | Adversarial Acc (%) | |
| --- | --- | --- | --- |
| | | FGSM | PGD[20] |
| Standard Training (Chen et al., 2020) | 91.5 | 25.6 | 0.8 |
| UAT (Jiang et al., 2020) | 66.6 | 26.2 | 21.4 |
| **UAT+UCR** (ours) | **72.0** | **30.7** | **24.6** |

**Result analysis.** As shown in Table 4, features learned by UAT is indeed more robust than standard training, *e.g.,* 0.8% to 21.4% under PGD attack. Moreover, with our proposed UCR regularizer (Eq. 26), we not only effectively improve both natural accuracy (66.6% to 72.0%), and also improve adversarial robustness: 26.2% to 30.7% under FGSM attack and 21.4% to 24.6% under PGD attack. This also helps justify the connection between contrastive learning and our CEM.

### A.3 CONCLUDING REMARK

With the above experiments on adversarial robustness, we empirically verify that our framework could serve as a valid probabilistic understanding of adversarial training and can be used to develop new effective adversarial training objectives.

## B OMITTED TECHNICAL DETAILS

For a concise presentation, we have omitted several technical details in the main text. Here we present a complete description of the derivation process.

### B.1 LOG LIKELIHOOD GRADIENT OF EBM

In Section 3, we have introduced Energy-based Models (EBM) and the gradient of their log likelihood. We now show how it can be derived out.

For a EBM in the following form,

$$p_{\boldsymbol{\theta}}(\mathbf{x}) = \frac{\exp\left(-E_{\boldsymbol{\theta}}(\mathbf{x})\right)}{Z(\boldsymbol{\theta})}, \tag{27}$$

the gradient of the log likelihood can be derived as follows:

$$\nabla_{\boldsymbol{\theta}} \mathbb{E}_{p_d(\mathbf{x})} \log p_{\boldsymbol{\theta}}(\mathbf{x}) \tag{28}$$

$$= -\mathbb{E}_{p_d(\mathbf{x})} \nabla_{\boldsymbol{\theta}} E_{\boldsymbol{\theta}}(\mathbf{x}) - \nabla_{\boldsymbol{\theta}} \log Z(\boldsymbol{\theta})$$

$$= \mathbb{E}_{p_d(\mathbf{x})} \nabla_{\boldsymbol{\theta}} E_{\boldsymbol{\theta}}(\mathbf{x}) + \frac{\nabla_{\boldsymbol{\theta}} \int_{\mathbf{x}} exp(-E_{\boldsymbol{\theta}}(\mathbf{x}))}{Z(\boldsymbol{\theta})}$$

$$= -\mathbb{E}_{p_d(\mathbf{x})} \nabla_{\boldsymbol{\theta}} E_{\boldsymbol{\theta}}(\mathbf{x}) + \int_{\hat{\mathbf{x}}} \frac{\exp(-E_{\boldsymbol{\theta}}(\hat{\mathbf{x}}))}{Z(\boldsymbol{\theta})} \nabla_{\boldsymbol{\theta}} E_{\boldsymbol{\theta}}(\hat{\mathbf{x}})$$

$$= \underbrace{-\mathbb{E}_{p_d(\mathbf{x})} \nabla_{\boldsymbol{\theta}} E_{\boldsymbol{\theta}}(\mathbf{x})}_{\text{positive phase}} + \underbrace{\mathbb{E}_{p_{\boldsymbol{\theta}}(\hat{\mathbf{x}})} \nabla_{\boldsymbol{\theta}} E_{\boldsymbol{\theta}}(\hat{\mathbf{x}})}_{\text{negative phase}}.$$

## B.2  CONNECTION BETWEEN STANDARD TRAINING AND JEM

In Section 4.4, we have claimed that if we replace the model distribution $p_{\boldsymbol{\theta}}(\hat{\mathbf{x}})$ with the data distribution $p_d(\mathbf{x})$ in Eqn. 10, the log likelihood gradient of JEM is equivalent to the negative gradient of the CE loss. Here we give a detailed proof as follows:

$$\nabla_{\boldsymbol{\theta}} \mathbb{E}_{p_d(\mathbf{x}, y)} \log p_{\boldsymbol{\theta}}(\mathbf{x}, y)$$

$$= \mathbb{E}_{p_d(\mathbf{x}, y)} \nabla_{\boldsymbol{\theta}} f_{\boldsymbol{\theta}}(\mathbf{x}, y) - \mathbb{E}_{p_{\boldsymbol{\theta}}(\hat{\mathbf{x}}) p_{\boldsymbol{\theta}}(\hat{y}|\hat{\mathbf{x}})} \nabla_{\boldsymbol{\theta}} f_{\boldsymbol{\theta}}(\hat{\mathbf{x}}, \hat{y})$$

$$\approx \mathbb{E}_{p_d(\mathbf{x}, y)} \nabla_{\boldsymbol{\theta}} f_{\boldsymbol{\theta}}(\mathbf{x}, y) - \mathbb{E}_{p_d(\mathbf{x}) p_{\boldsymbol{\theta}}(\hat{y}|\mathbf{x})} \nabla_{\boldsymbol{\theta}} f_{\boldsymbol{\theta}}(\mathbf{x}, \hat{y})$$

$$= \mathbb{E}_{p_d(\mathbf{x}, y)} \left[ \nabla_{\boldsymbol{\theta}} f_{\boldsymbol{\theta}}(\mathbf{x}, y) - \mathbb{E}_{p_{\boldsymbol{\theta}}(\hat{y}|\mathbf{x})} \nabla_{\boldsymbol{\theta}} f_{\boldsymbol{\theta}}(\mathbf{x}, \hat{y}) \right]$$

$$= \mathbb{E}_{p_d(\mathbf{x}, y)} \left[ \nabla_{\boldsymbol{\theta}} f_{\boldsymbol{\theta}}(\mathbf{x}, y) - \sum_{k=1}^{K} p_{\boldsymbol{\theta}}(k|\mathbf{x}) \nabla_{\boldsymbol{\theta}} f_{\boldsymbol{\theta}}(\mathbf{x}, k) \right]$$

$$= \mathbb{E}_{p_d(\mathbf{x}, y)} \left[ \nabla_{\boldsymbol{\theta}} f_{\boldsymbol{\theta}}(\mathbf{x}, y) - \sum_{k=1}^{K} \frac{\exp(f_{\boldsymbol{\theta}}(\mathbf{x}, k)) \nabla_{\boldsymbol{\theta}} f_{\boldsymbol{\theta}}(\mathbf{x}, k)}{\sum_{j=1}^{K} \exp(f_{\boldsymbol{\theta}}(\mathbf{x}, j))} \right]$$

$$= \mathbb{E}_{p_d(\mathbf{x}, y)} \left[ \nabla_{\boldsymbol{\theta}} f_{\boldsymbol{\theta}}(\mathbf{x}, y) - \sum_{k=1}^{K} \frac{\nabla_{\boldsymbol{\theta}} \exp(f_{\boldsymbol{\theta}}(\mathbf{x}, k))}{\sum_{j=1}^{K} \exp(f_{\boldsymbol{\theta}}(\mathbf{x}, j))} \right]$$

$$= \mathbb{E}_{p_d(\mathbf{x}, y)} \left[ \nabla_{\boldsymbol{\theta}} f_{\boldsymbol{\theta}}(\mathbf{x}, y) - \frac{\nabla_{\boldsymbol{\theta}} \sum_{k=1}^{K} \exp(f_{\boldsymbol{\theta}}(\mathbf{x}, k))}{\sum_{j=1}^{K} \exp(f_{\boldsymbol{\theta}}(\mathbf{x}, j))} \right]$$

$$= \mathbb{E}_{p_d(\mathbf{x}, y)} \left[ \nabla_{\boldsymbol{\theta}} f_{\boldsymbol{\theta}}(\mathbf{x}, y) - \nabla_{\boldsymbol{\theta}} \log \sum_{k=1}^{K} \exp(f_{\boldsymbol{\theta}}(\mathbf{x}, k)) \right]$$

$$= \mathbb{E}_{p_d(\mathbf{x}, y)} \nabla_{\boldsymbol{\theta}} \log \frac{\exp(f_{\boldsymbol{\theta}}(\mathbf{x}, y))}{\sum_{k=1}^{K} \exp(f_{\boldsymbol{\theta}}(\mathbf{x}, k))}$$

$$= \nabla_{\boldsymbol{\theta}} \mathbb{E}_{p_d(\mathbf{x}, y)} \log p_{\boldsymbol{\theta}}(y|\mathbf{x}). \tag{29}$$

## B.3  EQUIVALENCE BETWEEN AT LOSS AND CONTRASTIVE GRADIENT IN SUPERVISED LEARNING

In Section 4.3, we have claimed that the contrastive gradient equals to the negative gradient of the robust CE loss (AT loss) following the same deduction in Eqn. 29,

$$\mathbb{E}_{p_d(\mathbf{x}, y) \otimes p_{\boldsymbol{\theta}}(\hat{\mathbf{x}}, \hat{y})} \left[ \nabla_{\boldsymbol{\theta}} f_{\boldsymbol{\theta}}(\hat{\mathbf{x}}, y) - \nabla_{\boldsymbol{\theta}} f_{\boldsymbol{\theta}}(\hat{\mathbf{x}}, \hat{y}) \right]$$

$$= \mathbb{E}_{p_d(\mathbf{x}, y) \otimes p_{\boldsymbol{\theta}}(\hat{\mathbf{x}})} \left[ \nabla_{\boldsymbol{\theta}} f_{\boldsymbol{\theta}}(\hat{\mathbf{x}}, y) - \mathbb{E}_{p_{\boldsymbol{\theta}}(\hat{y}|\hat{\mathbf{x}})} \nabla_{\boldsymbol{\theta}} f_{\boldsymbol{\theta}}(\hat{\mathbf{x}}, \hat{y}) \right]$$

$$= \mathbb{E}_{p_d(\mathbf{x}, y)} \nabla_{\boldsymbol{\theta}} \log \frac{\exp(f_{\boldsymbol{\theta}}(\hat{\mathbf{x}}, y))}{\sum_{k=1}^{K} \exp(f_{\boldsymbol{\theta}}(\hat{\mathbf{x}}, k))}$$

$$= \mathbb{E}_{p_d(\mathbf{x}, y) \otimes p_{\boldsymbol{\theta}}(\hat{\mathbf{x}})} \nabla_{\boldsymbol{\theta}} \log p_{\boldsymbol{\theta}}(y|\hat{\mathbf{x}}), \tag{30}$$

which is exactly the negative gradient of the canonical AT loss (Madry et al., 2018).

### B.4 EQUIVALENCE BETWEEN INFONCE LOSS AND NON-PARAMETRIC CEM

In Section 6.1, we have claimed that the the log likelihood gradient of NP-CEM equals to exactly the negative gradient of the InfoNCE loss when we approximate $p_{\boldsymbol{\theta}}(\hat{\mathbf{x}})$ with $p_d(\hat{\mathbf{x}})$. The derivation is presented as follows:

$$
\begin{aligned}
&\mathbb{E}_{p_d(\mathbf{x},\mathbf{x}')}\nabla_{\boldsymbol{\theta}}f_{\boldsymbol{\theta}}(\mathbf{x},\mathbf{x}') - \mathbb{E}_{p_{\boldsymbol{\theta}}(\hat{\mathbf{x}},\hat{\mathbf{x}}')}\nabla_{\boldsymbol{\theta}}f_{\boldsymbol{\theta}}\left(\hat{\mathbf{x}},\hat{\mathbf{x}}'\right)\\
=&\mathbb{E}_{p_d(\mathbf{x},\mathbf{x}')}\nabla_{\boldsymbol{\theta}}f_{\boldsymbol{\theta}}(\mathbf{x},\mathbf{x}') - \mathbb{E}_{p_{\boldsymbol{\theta}}(\mathbf{x})}p_{\boldsymbol{\theta}}(\hat{\mathbf{x}}|\hat{\mathbf{x}}')\nabla_{\boldsymbol{\theta}}f_{\boldsymbol{\theta}}\left(\hat{\mathbf{x}},\hat{\mathbf{x}}'\right)\\
=&\mathbb{E}_{p_d(\mathbf{x},\mathbf{x}')}\nabla_{\boldsymbol{\theta}}f_{\boldsymbol{\theta}}(\mathbf{x},\mathbf{x}') - \mathbb{E}_{p_{\boldsymbol{\theta}}(\mathbf{x})}\frac{p_{\boldsymbol{\theta}}(\hat{\mathbf{x}},\hat{\mathbf{x}}')}{p_{\boldsymbol{\theta}}(\hat{\mathbf{x}}')}\nabla_{\boldsymbol{\theta}}f_{\boldsymbol{\theta}}\left(\hat{\mathbf{x}},\hat{\mathbf{x}}'\right)\\
=&\mathbb{E}_{p_d(\mathbf{x},\mathbf{x}')}\nabla_{\boldsymbol{\theta}}f_{\boldsymbol{\theta}}(\mathbf{x},\mathbf{x}') - \mathbb{E}_{p_{\boldsymbol{\theta}}(\hat{\mathbf{x}})}\int_{\hat{\mathbf{x}}}\frac{\exp(f_{\boldsymbol{\theta}}(\hat{\mathbf{x}},\hat{\mathbf{x}}'))}{\int_{\tilde{\mathbf{x}}}\exp(f_{\boldsymbol{\theta}}(\hat{\mathbf{x}},\tilde{\mathbf{x}}))}\nabla_{\boldsymbol{\theta}}f_{\boldsymbol{\theta}}(\hat{\mathbf{x}},\hat{\mathbf{x}}')\\
=&\mathbb{E}_{p_d(\mathbf{x},\mathbf{x}')}\nabla_{\boldsymbol{\theta}}f_{\boldsymbol{\theta}}(\mathbf{x},\mathbf{x}') - \mathbb{E}_{p_{\boldsymbol{\theta}}(\hat{\mathbf{x}})}\int_{\hat{\mathbf{x}}}\frac{p_d(\hat{\mathbf{x}})\exp(f_{\boldsymbol{\theta}}(\hat{\mathbf{x}},\hat{\mathbf{x}}'))}{\int_{\tilde{\mathbf{x}}}p_d(\tilde{\mathbf{x}})\exp(f_{\boldsymbol{\theta}}(\hat{\mathbf{x}},\tilde{\mathbf{x}}))}\nabla_{\boldsymbol{\theta}}f_{\boldsymbol{\theta}}(\hat{\mathbf{x}},\hat{\mathbf{x}}') \quad (\text{as } p_d(\hat{\mathbf{x}})=p_d(\tilde{\mathbf{x}})=\frac{1}{|\mathcal{X}|})\\
=&\mathbb{E}_{p_d(\mathbf{x},\mathbf{x}')}\nabla_{\boldsymbol{\theta}}f_{\boldsymbol{\theta}}(\mathbf{x},\mathbf{x}') - \mathbb{E}_{p_{\boldsymbol{\theta}}(\hat{\mathbf{x}})p_d(\hat{\mathbf{x}}')}\frac{\exp(f_{\boldsymbol{\theta}}(\hat{\mathbf{x}},\hat{\mathbf{x}}'))}{\mathbb{E}_{p_d(\tilde{\mathbf{x}})}\exp(f_{\boldsymbol{\theta}}(\hat{\mathbf{x}},\tilde{\mathbf{x}}))}\nabla_{\boldsymbol{\theta}}f_{\boldsymbol{\theta}}(\hat{\mathbf{x}},\hat{\mathbf{x}}')\\
\approx&\mathbb{E}_{p_d(\mathbf{x},\mathbf{x}')}\nabla_{\boldsymbol{\theta}}f_{\boldsymbol{\theta}}(\mathbf{x},\mathbf{x}') - \mathbb{E}_{p_d(\hat{\mathbf{x}})p_d(\hat{\mathbf{x}}')}\frac{\exp(f_{\boldsymbol{\theta}}(\hat{\mathbf{x}},\hat{\mathbf{x}}'))}{\mathbb{E}_{p_d(\tilde{\mathbf{x}})}\exp(f_{\boldsymbol{\theta}}(\hat{\mathbf{x}},\tilde{\mathbf{x}}))}\nabla_{\boldsymbol{\theta}}f_{\boldsymbol{\theta}}(\hat{\mathbf{x}},\hat{\mathbf{x}}') \quad\quad (31)\\
=&\mathbb{E}_{p_d(\mathbf{x},\mathbf{x}')}\nabla_{\boldsymbol{\theta}}f_{\boldsymbol{\theta}}(\mathbf{x},\mathbf{x}') - \mathbb{E}_{p_d(\hat{\mathbf{x}})}\nabla_{\boldsymbol{\theta}}\log\mathbb{E}_{p_d(\hat{\mathbf{x}}')}\exp(f_{\boldsymbol{\theta}}(\hat{\mathbf{x}},\hat{\mathbf{x}}'))\\
=&\mathbb{E}_{p_d(\mathbf{x},\mathbf{x}')}\left[\nabla_{\boldsymbol{\theta}}f_{\boldsymbol{\theta}}(\mathbf{x},\mathbf{x}') - \nabla_{\boldsymbol{\theta}}\log\mathbb{E}_{p_d(\hat{\mathbf{x}}')}\exp(f_{\boldsymbol{\theta}}(\hat{\mathbf{x}},\hat{\mathbf{x}}'))\right] \quad (\text{merge } p_d(\mathbf{x}) \text{ with } p_d(\hat{\mathbf{x}}))\\
=&\mathbb{E}_{p_d(\mathbf{x},\mathbf{x}')}\nabla_{\boldsymbol{\theta}}\log\frac{\exp(f_{\boldsymbol{\theta}}(\mathbf{x},\mathbf{x}'))}{\mathbb{E}_{p_d(\hat{\mathbf{x}}')}\exp(f_{\boldsymbol{\theta}}(\mathbf{x},\hat{\mathbf{x}}'))} \approx \frac{1}{N}\sum_{i=1}^{N}\nabla_{\boldsymbol{\theta}}\log\frac{\exp(f_{\boldsymbol{\theta}}(\mathbf{x}_i,\mathbf{x}_i'))}{\sum_{k=1}^{K}\exp(f_{\boldsymbol{\theta}}(\mathbf{x}_i,\hat{\mathbf{x}}_{ik}'))},
\end{aligned}
$$

where $(\mathbf{x},\mathbf{x}_i')$'s are positive samples from $p_d(\mathbf{x},\mathbf{x}')$ and $\hat{\mathbf{x}}_{ik}'$'s are negative samples independently drawn from $p_d(\hat{\mathbf{x}}')$.

### B.5 EQUIVALENCE BETWEEN ADVERSARIAL INFONCE AND NP-CEM

In Section 6.2, we have developed the unsupervised analogy of AT loss and regularization. In particular, we have claimed that contrastive gradient is equivalent to the gradient of the Adversarial InfoNCE loss (*i.e.,* the InfoNCE loss of the adversarial example $\hat{\mathbf{x}}$) utilized in previous work (Jiang et al., 2020; Ho & Vasconcelos, 2020; Kim et al., 2020). It can be derived following Eqn. 31:

$$
\begin{aligned}
&\mathbb{E}_{p_d(\mathbf{x},\mathbf{x}')\otimes p_{\boldsymbol{\theta}}(\hat{\mathbf{x}},\hat{\mathbf{x}}')}\left[\nabla_{\boldsymbol{\theta}}f_{\boldsymbol{\theta}}(\hat{\mathbf{x}},\mathbf{x}')-\nabla_{\boldsymbol{\theta}}f_{\boldsymbol{\theta}}(\hat{\mathbf{x}},\hat{\mathbf{x}}')\right]\\
=&\mathbb{E}_{p_d(\mathbf{x},\mathbf{x}')\otimes p_{\boldsymbol{\theta}}(\hat{\mathbf{x}})}\left[\nabla_{\boldsymbol{\theta}}f_{\boldsymbol{\theta}}(\hat{\mathbf{x}},\mathbf{x}') - \mathbb{E}_{p_{\boldsymbol{\theta}}(\hat{\mathbf{x}}'|\hat{\mathbf{x}})}\nabla_{\boldsymbol{\theta}}f_{\boldsymbol{\theta}}(\hat{\mathbf{x}},\hat{\mathbf{x}}')\right]\\
=&\mathbb{E}_{p_d(\mathbf{x},\mathbf{x}')\otimes p_{\boldsymbol{\theta}}(\hat{\mathbf{x}})}\left[\nabla_{\boldsymbol{\theta}}f_{\boldsymbol{\theta}}(\hat{\mathbf{x}},\mathbf{x}') - \mathbb{E}_{p_d(\hat{\mathbf{x}}')}\frac{\exp(f_{\boldsymbol{\theta}}(\hat{\mathbf{x}},\hat{\mathbf{x}}'))}{\mathbb{E}_{p_d(\tilde{\mathbf{x}})}\exp(f_{\boldsymbol{\theta}}(\hat{\mathbf{x}},\hat{\mathbf{x}}'))}\nabla_{\boldsymbol{\theta}}f_{\boldsymbol{\theta}}(\hat{\mathbf{x}},\hat{\mathbf{x}}')\right]\\
=&\mathbb{E}_{p_d(\mathbf{x},\mathbf{x}')\otimes p_{\boldsymbol{\theta}}(\hat{\mathbf{x}})}\nabla_{\boldsymbol{\theta}}\log\frac{\exp(f_{\boldsymbol{\theta}}(\hat{\mathbf{x}},\mathbf{x}'))}{\mathbb{E}_{p_d(\hat{\mathbf{x}}')}\exp(f_{\boldsymbol{\theta}}(\hat{\mathbf{x}},\hat{\mathbf{x}}')}\\
\approx&\frac{1}{N}\sum_{i=1}^{N}\nabla_{\boldsymbol{\theta}}\log\frac{\exp(f_{\boldsymbol{\theta}}(\hat{\mathbf{x}}_i,\mathbf{x}_i'))}{\sum_{k=1}^{K}\exp(f_{\boldsymbol{\theta}}(\hat{\mathbf{x}}_i,\hat{\mathbf{x}}_{ik}'))}, \quad\quad (32)
\end{aligned}
$$

where $(\mathbf{x}_i,\mathbf{x}_i')$ are positive samples drawn from $p_d(\mathbf{x},\mathbf{x}')$, $\hat{\mathbf{x}}_i$'s are adversarial samples drawn from $p_{\boldsymbol{\theta}}(\hat{\mathbf{x}})$, and $\hat{\mathbf{x}}_{ik}'$'s are negative samples independently drawn from $p_d(\hat{\mathbf{x}}')$.

## C MORE DETAILS AND RESULTS ON ADVERSARIAL SAMPLING

### C.1 DETAILED EXPERIMENTAL SETUP

**Supervised Adversarial Sampling.** For supervised robust models, we adopt the same pretrained ResNet50 checkpoint on CIFAR-10 as Santurkar et al. (2019) [3] for a fair comparison. As described

---

[3]We download the checkpoint from the repository `https://github.com/MadryLab/robustness_applications`.

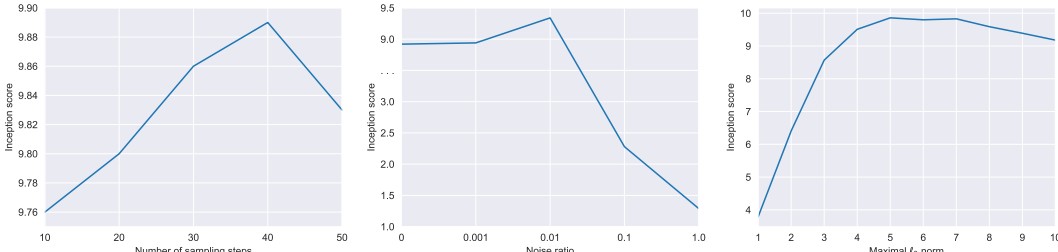

Figure 2: Algorithmic analysis of our proposed supervised adversarial sampling algorithm (RCS). Left: Inception score with increasing sampling steps $N$. Middle: Inception score with increasing diffusion noise scale. Right: Inception score with increasing $\ell_2$-norm bound $\beta$.

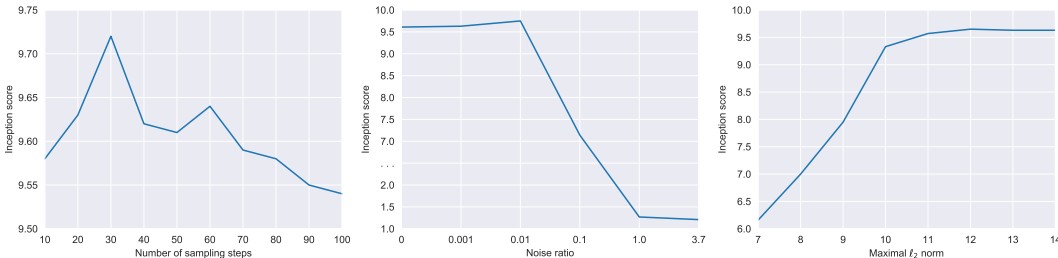

Figure 3: Algorithmic analysis of our proposed unsupervised adversarial sampling algorithm (MaxEnt). Left: Inception score with increasing sampling steps $N$. Middle: Inception score with increasing diffusion noise scale. Right: Inception score with increasing $\ell_2$-norm bound $\beta$.

in Santurkar et al. (2019), the ResNet-50 model is adversarially trained for 350 epochs with learning rate 0.01 and batch size 256. The learning rate is dropped by 10 at epoch 150 and 200. The training attack is $\ell_2$-norm PGD attack with random start, maximal perturbation norm 0.5, step size 0.1 and 7 steps.

**Unsupervised Adversarial Sampling.** As far as we know, we are the first to consider sampling from unsupervised robust models. Therefore, to obtain an unsupervised robust model for sampling, we adopt ACL (Jiang et al., 2020) as the baseline method and follow their default hyper-parameters [4] to train it. The official implementation of ACL is built upon the SimCLR (Chen et al., 2020) framework for contrastive learning, while they choose a specific hyper-parameter configuration for adversarial training. In particular, they choose 512 for batch size and train for 1000 epochs with ResNet-18 backbone (He et al., 2016). The base learning rate is 1.0, where they use a linear warm up strategy for the first 10 epochs, and apply cosine annealing scheduler after that. The training attack is kept the same as that of the supervised case for a fair comparison. For ResNet-18, we adopt the ACL(A2A) setting, which adopts the normal ResNet with only one Batch Normalization module. Instead, for ResNet-50, we notice that the ACL(A2S) setting yields slightly better results. The ResNet variants in the ACL(A2S) setting contains two BN modules, where we assign natural and adversarial examples to different modules. We refer more details to the original paper (Jiang et al., 2020).

**Evaluation of sample quality.** Note that there are four hyper-parameters in our sampling protocol: step size $\alpha$, $\ell_2$-ball size $\beta$, noise scale $\eta$, and iteration steps $K$, for which we list our choice in Table 5. We evaluate sample quality quantitatively *w.r.t.* the Inception Score (IS) (Salimans et al., 2016) and Fréchet Inception Distance (FID) (Heusel et al., 2017) with 50,000 samples, where the standard variation of IS is around 0.1.

## C.2 Additional Analysis

Besides the results presented in the main text, we also conduct more experiments to analyze the behavior of our proposed adversarial sampling algorithms, both quantitatively and qualitatively. We conduct a detailed analysis of our proposed sampling algorithms and present the results of supervised

---

[4]Official code: `https://github.com/VITA-Group/Adversarial-Contrastive-Learning`.

Table 5: Sampling hyper-parameters in each scenario.

| Scenario | Model | $\alpha$ | $\beta$ | $\eta$ | $K$ |
|---|---|---|---|---|---|
| Supervised | ResNet50 | 1 | 6 | 0.01 | 20 |
| Unsupervised | ResNet18 | 7 | $\infty$ | 0.0 | 10 |
| | ResNet50 | 7 | $\infty$ | 0.0 | 50 |

adversarial sampling (with RCS) in Figure 2 and the results of unsupervised adversarial sampling (with MaxEnt) in Figure 3. Note that in both cases, we adopt the ResNet-50 backbone and use the default hyper-parameters unless specified.

### C.2.1  CHAIN LENGTH

From the left panels of Figure 2 and Figure 3, we can see the two adversarial algorithms both have a sweet spot of sampling steps $N$ (length of the sampling Markov chains) at around 30 to 40 steps, before and after which the results will be slightly worse.

### C.2.2  NOISE RATIO

In the proper Langevin dynamics, the scale of the noise is determined by the step size, $\eta = \sqrt{2\alpha}$. However, in practice, this would lead to a catastrophically degraded sample quality as the noise takes over the gradient information. Therefore, following Song & Ermon (2019) and Grathwohl et al. (2019), we also anneal the noise ratio $\eta$ to a smaller value for better sample quality. As shown in the middle panels of Figure 2 and Figure 3, the optimal noise ratio is around 0.01 for both cases.

### C.2.3  MAXIMAL NORM

An apparent difference of our adversarial sampling algorithms to the canonical Langevin dynamics is that ours have a projection step that limits the distance between the samples and the initial seeds. In the right panels of Figure 2 and Figure 3, we show the impact of the scale of the $\ell_2$-norm ball for the sample quality. We can see that generally speaking, as the ball grows larger, the samples get refined. In the supervised case, the sample quality gets slightly worse with a large norm, which does not happen in the unsupervised case.

### C.2.4  SAMPLING TRAJECTORY

Aside from the qualitative inspection of the proposed sampling algorithms, we also demonstrate the sampling trajectory of our supervised (RCS) and unsupervised (MaxEnt) adversarial sampling methods in Figure 4 & 5. We can see that the samples get gradually refined in terms both low-level textures and high-level semantics.

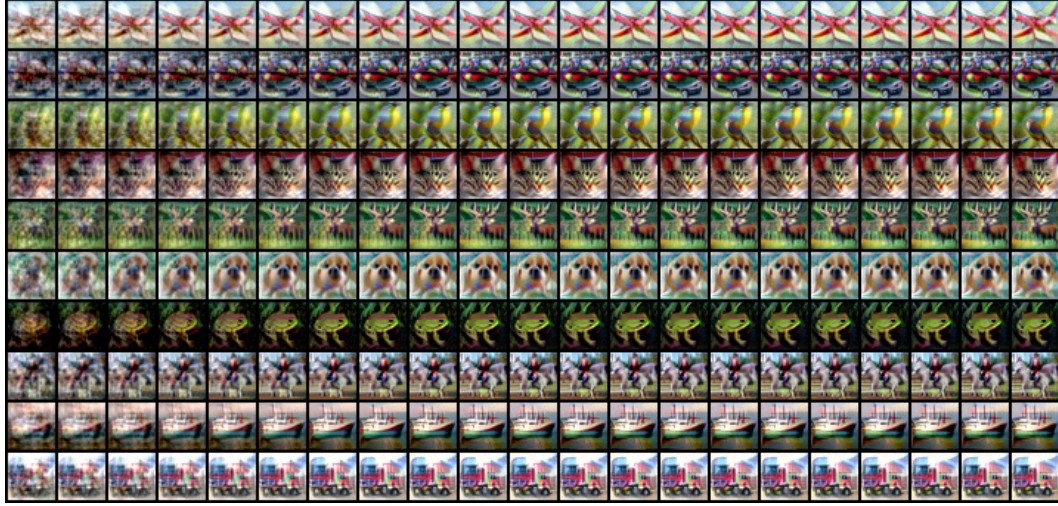

Figure 4: Sampling trajectory (the first 20 steps) of our proposed supervised adversarial sampling algorithm (RCS). Each row represents the refinement progress of a single sample.

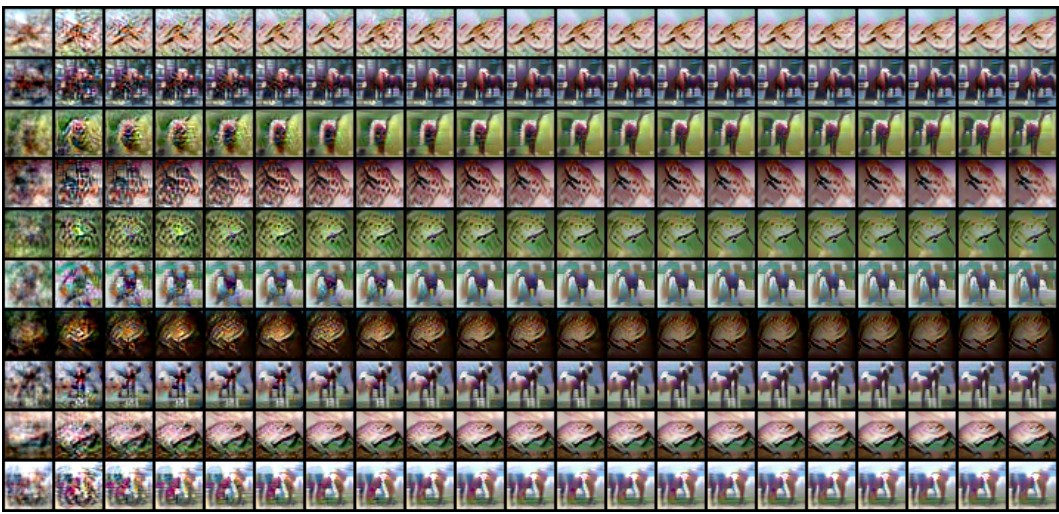

Figure 5: Sampling trajectory (the first 20 steps) of our proposed unsupervised adversarial sampling algorithm (MaxEnt). Each row represents the refinement progress of a single sample.

