# OpenReview forum: "A Unified Contrastive Energy-based Model for Understanding the Generative Ability of Adversarial Training"
_ICLR.cc/2022/Conference — ICLR 2022 Poster_

### Official Review · Reviewer_9aEk · 2021-11-02

**Correctness:** 4
**Technical Novelty And Significance:** 4
**Empirical Novelty And Significance:** 4
**Recommendation:** 8
**Confidence:** 4

**Main Review:**

In general, I like the direction in this paper. By formulating it as an energy-based model, we can look more into the normalization term of the softmax which is quite often overlooked. The connection drawn between Langevin dynamics and adversarial training is nice and new at least to me. I think the insight in contrastive learning is quite interesting. Although the so-called "hard negative mining" is an old idea in the computer vision community, it's not generally been casted in a theoretical framework. I enjoyed reading the paper which is not quite often (anymore) for my review batches in the ML conferences.

There is this small confusion in the notation that p_theta(x,y) defined in eq. (2) seems to be used equivalently with p_theta(y|x), it's better to clarify which part of f_theta(x,y) was attributed to the prior p(x), or my understanding would be eq.(2) should rather be a definition of p_theta(y|x) (where it doesn't take into account the prior).

I'm also a little bit confused initially about the sign between eq. (12) and eq. (7). In eq.(7), the goal for the minimization step is MLE, which means to minimize - logp (or equivalently, maximize log p). In eq. (12), we are talking about the AT loss being the same as negative gradient of the loss in eq.(7), which is correct, but in order to match the sign w.r.t. the MLE in eq. (7), seems that we should be running gradient ascent. This is clear in the end from the appendix, but it might worth mentioning a single word in the main paper as well.

The results are also very competitive, I think the unsupervised contrastive results are especially quite strong. It would be nice to have some additional technical explanations why the methods of moments approaches would work better than this current approach, rather than only a descriptive explanation in the current manuscript (although that is nice to have as well).

In terms of the results, it is interesting to see unsupervised Langevin performing better than supervised Langevin method. Any insights in that?

Although the paper focused on generative models, I also would be wondering whether the importance sampling approach might have consequences for discriminative models as well (the more common application in computer vision). It might be nice to show that as well since it shouldn't be a lot of work.

Nvasconcelos should be Vasconcelos.

**Summary Of The Paper:**

By combining generative models and discriminative models in the framework of an energy-based model, this paper provides several theoretical insights about adversarial training, including that the softmax normalization term already provides a hint on adversarial training, and the importance sampling (with a justification) approach to unsupervised contrastive learning. The experiments show that their strategy with Langevin sampling improves over simpler adversarial training strategies.


**Summary Of The Review:**

Overall, I like the insights provided in the paper and their experiments also proved their points well. Solid accept.

---

> ### Author Response · Authors · 2021-11-18
> **Response to Reviewer 9aEk (2/2)**
>
> **Q4.** Why does unsupervised Langevin perform better than the supervised Langevin method?
>
> **A4.** Indeed the unsupervised results are surprisingly good, nevertheless, we have to note that the two (supervised and unsupervised Langevin) are actually comparable and the winner depends on the chosen metric (IS or FID). And we believe that why unsupervised CEM could be on par with the supervised CEM might be contributed to the aggressive data augmentations in contrastive learning. These augmentations, including RandomResizedCrop, ColorJitter, Grayscale, flip, etc, incorporate **many variants of the input data**, which might be not that helpful (even harmful) for the classification when labels are available, but could instead be very helpful for modelling the input distribution for the unsupervised CEM. On the other hand, the supervised CEM could benefit from the label information that complements the lack of the augmentations. As a result, the two might reach a tie. We will dig deeper into this relationship in the future.
>
> ---
> **Q5.** Does the importance sampling approach have consequences for discriminative models as well?
>
> **A5.** Yes! As stated above, our CEM is a joint model, so it would also be interesting to study its discriminative part, although it is not the main focus of this paper. In our analysis, the importance sampling of CEM corresponds to the process of generating adversarial examples, so in this part, we evaluate the possible implication of our CEM framework on adversarial robustness, i.e., the discriminative ability on adversarial data.
>
> Therefore, besides the adversarial sampling experiments, in the revised version, we add a whole new section **Appendix D**, where we show that new regularization terms, CR (supervised) and UCR (unsupervised), derived from our CEM framework could also bring significant improvement of robustness in both supervised and unsupervised scenarios. We list the main results below.
>
> #### i) Supervised AT on CIFAR-10 with PGD attack.
>
> | Model        | Training                              | Natural Acc (\%) | Adversarial Acc (\%) |
> |--------------|---------------------------------------|:----------------:|:--------------------:|
> | ResNet18     | AT (baseline)           |   **83.7**  |         52.2         |
> |              | TRADES  |      82.5      |         54.3         |
> |              | **AT+CR (ours)**                 |       81.5       |     **55.2**    |
> | WideResNet34 | AT (baseline)         |   **86.8**  |         53.6         |
> |              | TRADES  |      83.4      |     **57.0**    |
> |              | **AT+CR (ours)**                |       86.6       |     **57.0**   |
>
> #### ii)  Unsupervised AT on CIFAR-10 with ResNet-18 backbone, FGSM and PGD attack.
>
> | Training                     | Natural Acc  (\%) | Adversarial Acc  (\%) (FGSM)  | Adversarial Acc  (\%) (PGD)           |
> |------------------------------|:-----------------:|:--------------------:|:----------:|
> | Standard Training (baseline) |        91.5       |         25.6         |     0.8    |
> | UAT (baseline)               |        66.6       |         26.2         |    21.4    |
> | **UAT+UCR (ours)**               |        72.0       |         30.7         |    24.6    |
>
> From  the results above, we can conclude that:
> - For supervised AT, we can see that our proposed  AT+CR objective derived from our framework indeed enjoys much better robustness than vanilla AT on **two different models, ResNet18 and WideResNet34**. Compared to the state-of-the-art AT variant TRADES, we can see that AT+CR is comparable, and sometimes slightly better at robustness. When the two have similar robustness (for WideResNet34), AT+CR obtains better natural accuracy than TRADES (86.6 v.s. 83.4). These results empirically justify that our interpretation of AT and TRADES from a probabilistic perspective.
> - For unsupervised AT, similarly, we can see that unsupervised AT can indeed improve the model robustness by a large margin. Moreover,
> - our proposed UCR regularization not only improves the natural accuracy of UAT (66.6$\to$72.0) but also improve its robustness under **either FGSM or PGD attack**.
>
> Therefore, our CEM framework that interprets AT from a probabilistic framework can also be used to devise new AT methods and improve model robustness.
>
>
> ---
> Thanks for your detailed and constructive review. Hope our explanations could address your concerns! Please let us know if you have additional questions.

---

> > ### Comment · Reviewer_9aEk · 2021-11-28
> > **Thanks for the rebuttal!**
> >
> > I would like to thank the authors to write a comprehensive rebuttal that cleared up all my questions. I have liked this paper and still like it right now. It's nice to see some of the results on discriminative models too. I'm still strongly supporting acceptance.

---

> ### Author Response · Authors · 2021-11-18
> **Response to Reviewer 9aEk (1/2)**
>
>
> We thank Reviewer 9aEk for careful reading of our work and appreciating its theoretical insights. We will address your concerns in the following points.
>
> ---
>
> **Q1.** Confusion between $p_\theta(x,y)$  and $p_\theta(y|x)$.
> > A small confusion in the notation that $p_\theta(x,y)$ defined in Eq. (2) seems to be used equivalently with $p_\theta(y|x)$. It's better to clarify which part of $f_\theta(x,y)$ was attributed to the prior $p(x)$, or my understanding would be eq.(2) should rather be a definition of p_theta(y|x) (where it doesn't take into account the prior).
>
> **A1.** Here $p_\theta(x,y)={\exp(f(x,y))/Z(\theta)}$ in Eq. (2) defines the **joint distribution** between $x$ and $y$, based on which, we can easily **derive** the marginal $p(x)$ and the conditional $p(y|x)$ as:
> $$p(x)=\sum_y p(x,k)=\frac{\sum_y\exp(f(x, k))}{Z(\theta)}, \quad p(y|x)=\frac{p(x,y)}{p(x)}=\frac{\exp(f(x, y))/Z(\theta)}{\left(\sum_k\exp(f(x, k))\right)/Z(\theta)}=\frac{\exp(f(x, y))}{\sum_k\exp(f(x, k))}.$$
> The former is the prior $p(x)$ (generative part) that we use for Langevin sampling, and the latter is the softmax scores used in the CE loss (discriminative part),  making our CEM become a *joint model*. The main difference between $p(x,y)$ and $p(y|x)$ is the denominator, either over all possible inputs $Z(\theta)=\int_x\exp(f(x,y))$ or merely over the prediction scores $\sum_{k}\exp(f(x,k))$ of a given input $x$. Notably, the derivation above holds for general similarity function $f_\theta(x,y)$ taking any form, so the specific form of $f_\theta(x,y)$ does not matter here.
>
>
> ---
> **Q2.** A little bit confused initially about the sign between eq. (12) and eq. (7) ... This is clear in the end from the appendix, but it might worth mentioning a single word in the main paper as well.
>
> **A2.** Thanks for your suggestion and we have explained it in the revised version (below Eq. 12).
>
> ---
> **Q3.**  It would be nice to have some additional technical explanations why the methods of moments approaches would work better than this current approach.
>
> **A3.** We guess that by methods of moments you are referring to the Score Matching (SM) method, which actually has a large difference to our method that learns an energy-based model (CEM) with adversarial training (AT). In particular, the two methods have
> 1) different training method: image denoising (SM) v.s. minimax optimization (AT);
> 2) different models: EBMs model $f:\mathbb{R}^N\to\mathbb{R}$ while SM models $g:\mathbb{R}^N\to\mathbb{R}^N$, as result, they adopt very different architectures, ResNet (CEM) and U-Net (SN).
>
> Therefore, it might be hard to directly compare and explain the performance of the two methods. Overall we believe that both methods are promising probabilistic methods to explore. Among other attempts, our work points out a new promising direction, adversarial training, with both stable training and good performance for EBM training.

---

### Official Review · Reviewer_rJdV · 2021-11-02

**Correctness:** 3
**Technical Novelty And Significance:** 2
**Empirical Novelty And Significance:** 2
**Recommendation:** 5
**Confidence:** 3

**Main Review:**

The contribution is a unified probabilistic framework to explain adversarial learning, however, it is not clear whether this probabilistic framework can be beneficial for improving the training of adversarial models.

The sampling strategy is not claimed as a contribution of the proposed work, however, the experiment part spends a lot space to demonstrate that the sampling in the proposed framework is better than existing works. The authors need to more clearly elaborate the contributions.

The proposed probabilistic framework is generic, and thus should be able to work with different large-scale and large-resolution datasets. The authors only show results on a relatively small-scale and low-resolution dataset. The results presented are not ideal to explain the effectiveness of the proposed framework.





**Summary Of The Paper:**

The submission proposes a unified probabilistic framework to illustrate the generative capability of adversarial training. It also offers a unified perspective of adversarial training and sampling from both a supervised learning setting and an unsupervised learning setting. The proposed adversarial sampling strategy from the method is extensively demonstrated on different benchmarks, showing better quality compared with existing related works.

**Summary Of The Review:**

The problem studied is interesting, however, more investigation should be conducted to show the effectiveness and importance of the proposed probabilistic framework. Please also check the weakness part for more explanation.

---

> ### Author Response · Authors · 2021-11-18
> **Response to Reviewer rJdV (2/2)**
>
> **Q3.** The proposed probabilistic framework is generic and thus should be able to work with different large-scale and large-resolution datasets.
>
> **A3.** Indeed, it is better to evaluate it on a large-scale dataset like ImageNet. Unfortunately, the bottleneck here is the computation cost of Adversarial Training (AT), which is very high because of the process of generating adversarial samples in each update, making it typically 10x slower than standard training. Therefore, it is hardly affordable to adversarially train a model on ImageNet. In fact, almost all state-of-the-art AT methods only conduct experiments on medium-scale & low-resolution datasets like MNIST, CIFAR-10, and CIFAR-100 [1,2,3,4,5]. In the paper, as we mainly focus on the probabilistic interpretation of adversarial training (instead of developing fast AT methods), we also follow the standard AT protocol for a fair analysis.
>
> To further verify the effectiveness of our CEM, following your suggestions, we additionally experiment on **CIFAR-100** with more classes, and the results are shown below. We only evaluate Inception Score (IS) due to the lack of public CIFAR-100 statistics in the [official code of FID](https://github.com/bioinf-jku/TTUR).
>
> | Training |  Sampling  | Method | IS (↑) |
> | ---| ---| ---| ---|
> | Supervised | Conditional | TA (Santurkar et al., 2019) |  4.85
> || | Langevin (ours) | **8.15**  |
> || | CS (ours) | 7.54 |
> || | RCS (ours) | 7.78 |
> |Unsupervised| Conditional | PGD |  7.36 |
> || | Langevin (ours) | **7.67** |
> || Unconditional | PGD | 4.33 |
> || | Langevin (ours) | **6.16** |
>
> As CIFAR-100 has more classes than CIFAR-10, the IS scores are a little worse than CIFAR-10. Nevertheless, we can still see that our proposed sampling algorithms have clear advantages over TA proposed by Santurkar et al. Meanwhile, the unsupervised models also obtain relatively good sample quality.
>
> Ref:
>
> [1] Zhang et al. Theoretically principled trade-off between robustness and accuracy. ICML 2019.
>
> [2] Wu and Wang. Adversarial Weight Perturbation Helps Robust Generalization. NeurIPS 2020.
>
> [3] Rebuffi et al. Fixing Data Augmentation to Improve Adversarial Robustness. arxiv preprint 2021.
>
> [4] Jiang et al.  Robust pre-training by adversarial contrastive learning. NeurIPS 2020.
>
> [5] Kim et al. Adversarial Self-Supervised Contrastive Learning. NeurIPS 2020.
>
> ---
> Thanks for your review and hope our revisions and extended results could address your concerns. We are willing to address your further questions before the discussion stage closes.

---

> ### Author Response · Authors · 2021-11-18
> **Response to Reviewer rJdV (1/2)**
>
> We thank Reviewer rJdV for appreciating the novelty and insights of our work. We will address your concerns as follows.
>
> ---
> **Q1.** It is not clear whether this probabilistic framework can be beneficial for improving the training of adversarial models.
>
> **A1.** Yes, it could. Although our paper mainly focuses on explaining the generative ability of AT, in the revised version, we add a new section **Appendix D**, that our CEM framework could also inspire effective objectives for improving adversarial robustness. Specifically, we show that new regularization terms, CR (supervised) and UCR (unsupervised), derived from our CEM framework could also bring significant improvement of robustness in both supervised and unsupervised scenarios. We list the main results below.
>
> #### i) Supervised AT on CIFAR-10 with PGD attack.
>
> | Model        | Training                              | Natural Acc (\%) | Adversarial Acc (\%) |
> |--------------|---------------------------------------|:----------------:|:--------------------:|
> | ResNet18     | AT (baseline)           |   **83.7**  |         52.2         |
> |              | TRADES  |      82.5      |         54.3         |
> |              | **AT+CR (ours)**                 |       81.5       |     **55.2**    |
> | WideResNet34 | AT (baseline)         |   **86.8**  |         53.6         |
> |              | TRADES  |      83.4      |     **57.0**    |
> |              | **AT+CR (ours)**                |       86.6       |     **57.0**   |
>
> #### ii)  Unsupervised AT on CIFAR-10 with ResNet-18 backbone, FGSM and PGD attack.
>
> | Training                     | Natural Acc  (\%) | Adversarial Acc  (\%) (FGSM)  | Adversarial Acc  (\%) (PGD)           |
> |------------------------------|:-----------------:|:--------------------:|:----------:|
> | Standard Training (baseline) |        91.5       |         25.6         |     0.8    |
> | UAT (baseline)               |        66.6       |         26.2         |    21.4    |
> | **UAT+UCR (ours)**               |        72.0       |         30.7         |    24.6    |
>
> From  the results above, we can conclude that:
> - For supervised AT, we can see that our proposed  AT+CR objective derived from our framework indeed enjoys much better robustness than vanilla AT on **two different models, ResNet18 and WideResNet34**. Compared to the state-of-the-art AT variant TRADES, we can see that AT+CR is comparable, and sometimes slightly better at robustness. When the two have similar robustness (for WideResNet34), AT+CR obtains better natural accuracy than TRADES (86.6 v.s. 83.4). These results empirically justify that our interpretation of AT and TRADES from a probabilistic perspective.
> - For unsupervised AT, similarly, we can see that unsupervised AT can indeed improve the model robustness by a large margin. Moreover, our proposed UCR regularization not only improves the natural accuracy of UAT (66.6$\to$72.0) but also improve its robustness under **either FGSM or PGD attack**.
>
> Therefore, our CEM framework that interprets AT from a probabilistic framework can also be used to devise new AT methods and improve model robustness.
>
> ---
> **Q2.** The sampling strategy is not claimed as a contribution of the proposed work,  and the authors need to more clearly elaborate the contributions.
>
> **A2.** Thanks for pointing it out. In the summarized contributions in **Section 1** (Point 1 & 3), we have mentioned that our framework could inspire new adversarial sampling methods. Following your suggestions, we have made them more clear by adding them to the heading of each point in the revised version.

---

> ### Author Response · Authors · 2021-11-22
> **Need further clarification?**
>
> Thanks for your constructive comments. We have tried our best to address the concerns. Is there any unclear point that we should/could further clarify?

---

> ### Author Response · Authors · 2021-11-28
> **Please give your further opinions on our paper**
>
> Dear Reviewer rJdV,
>
> We have updated our manuscript and replied to your comments. Would you please check whether our efforts are satisfactory and raise your score? Many thanks!
>
> Authors

---

### Official Review · Reviewer_MiyJ · 2021-11-03

**Correctness:** 4
**Technical Novelty And Significance:** 3
**Empirical Novelty And Significance:** 3
**Recommendation:** 6
**Confidence:** 2

**Main Review:**

This paper has 4 major contributions.
1. The proposed Contrastive Energy-based Models (CEM) gives a probabilistic interpretation for adversarial training, which explains the generative ability of adversarial trained models.
2. CEM's parametric and non-parametric form give probabilistic understanding of previous ST and AT in both supervised and unsupervised settings.
3. Under CEM, the equivalence between the IS and the InfoNCE loss of contrastive learning is established, which enables us to design principled adversarial sampling for unsupervised learning.
4. Experiments show that adversarial sampling methods derived under CEM outperforms or achieve comparable performance to SOTA.

However, more experiments using different models, different dataset and different attacks are needed.

**Summary Of The Paper:**

This paper proposed a unified probabilistic framework, dubbed as Contrastive Energy-based Models, to understand the robustness and generative capability. The proposed CEM is a special case of EBM that models the joint distribution over two variables with a similarity function defined in a contrastive form. CEM could be instantiated into parametric form and non-parametric form, which work for supervised learning and unsupervised learning respectively. CEM could demystify adversarial training's generative capability in both supervised and unsupervised setting. Moreover, with CEM, adversarial training could be extended to unsupervised scenario.

**Summary Of The Review:**

The proposed unified probabilistic framework CEM is fundamentally novel and interesting. It gives explanation on the generative capability of adversarial trained model. It also gives probabilistic understanding of ST and AT in both supervised and unsupervised learnings. The derived adversarial sampling method show a better sample quality. However, more experiments are needed to further justify the correctness of the proposed CEM, given that there are a lot of approximation in the derivations.

---

> ### Author Response · Authors · 2021-11-18
> **Response to Reviewer MiyJ**
>
> We thank Reviewer MiyJ for appreciating the novelty and insights of our work. Following your suggestions, we have more experiments with different models, datasets, and attacks for a thorough justification of our CEM model and our analysis of adversarial training.
>
> ### a) Image Generation on CIFAR-100
>
> Besides the image generation results on CIFAR-10, here we also provide a study of the adversarial sampling algorithms on **a different dataset, CIFAR-100**, another commonly used dataset for adversarial training [1,2]. Experiments are conducted with ResNet18 for efficiency with the same training protocol as for CIFAR-10 (Sec 6.1). We only evaluate Inception Score (IS) due to the lack of public CIFAR-100 statistics in the [official code of FID](https://github.com/bioinf-jku/TTUR).
>
> | Training |  Sampling  | Method | IS (↑) |
> | ---| ---| ---| ---|
> | Supervised | Conditional | TA (Santurkar et al., 2019) |  4.85
> || | Langevin (ours) | **8.15**  |
> || | CS (ours) | 7.54 |
> || | RCS (ours) | 7.78 |
> |Unsupervised| Conditional | PGD |  7.36 |
> || | Langevin (ours) | **7.67** |
> || Unconditional | PGD | 4.33 |
> || | Langevin (ours) | **6.16** |
>
> As CIFAR-100 has more classes than CIFAR-10, the IS scores are a little worse than CIFAR-10. Nevertheless, we can still see that our proposed sampling algorithms have clear advantages over TA proposed by Santurkar et al. Meanwhile, the unsupervised models also obtain relatively good sample quality.
>
> Ref:
>
> [1] Wu and Wang. Adversarial Weight Perturbation Helps Robust Generalization. NeurIPS 2020.
>
> [2] Rebuffi et al. Fixing Data Augmentation to Improve Adversarial Robustness. arxiv preprint 2021.
>
> ### b) Adversarial Robustness
>
> Although our paper mainly focuses on explaining the generative ability of AT, in the revised version, we add a new section **Appendix D**, that our CEM framework could also inspire effective objectives for improving adversarial robustness. Specifically, we show that new regularization terms, CR (supervised) and UCR (unsupervised), derived from our CEM framework could also bring significant improvement of robustness in both supervised and unsupervised scenarios. We list the main results below.
>
> #### i) Supervised AT on CIFAR-10 with PGD attack.
>
> | Model        | Training                              | Natural Acc (\%) | Adversarial Acc (\%) |
> |--------------|---------------------------------------|:----------------:|:--------------------:|
> | ResNet18     | AT (baseline)           |   **83.7**  |         52.2         |
> |              | TRADES  |      82.5      |         54.3         |
> |              | **AT+CR (ours)**                 |       81.5       |     **55.2**    |
> | WideResNet34 | AT (baseline)         |   **86.8**  |         53.6         |
> |              | TRADES  |      83.4      |     **57.0**    |
> |              | **AT+CR (ours)**                |       86.6       |     **57.0**   |
>
> #### ii)  Unsupervised AT on CIFAR-10 with ResNet-18 backbone, FGSM and PGD attack.
>
> | Training                     | Natural Acc  (\%) | Adversarial Acc  (\%) (FGSM)  | Adversarial Acc  (\%) (PGD)           |
> |------------------------------|:-----------------:|:--------------------:|:----------:|
> | Standard Training (baseline) |        91.5       |         25.6         |     0.8    |
> | UAT (baseline)               |        66.6       |         26.2         |    21.4    |
> | **UAT+UCR (ours)**               |        72.0       |         30.7         |    24.6    |
>
> From  the results above, we can conclude that:
> - For supervised AT, we can see that our proposed  AT+CR objective derived from our framework indeed enjoys much better robustness than vanilla AT on **two different models, ResNet18 and WideResNet34**. Compared to the state-of-the-art AT variant TRADES, we can see that AT+CR is comparable, and sometimes slightly better at robustness. When the two have similar robustness (for WideResNet34), AT+CR obtains better natural accuracy than TRADES (86.6 v.s. 83.4). These results empirically justify that our interpretation of AT and TRADES from a probabilistic perspective.
> - For unsupervised AT, similarly, we can see that unsupervised AT can indeed improve the model robustness by a large margin. Moreover, our proposed UCR regularization not only improves the natural accuracy of UAT (66.6$\to$72.0) but also improve its robustness under **either FGSM or PGD attack**.
>
> Therefore, our CEM framework that interprets AT from a probabilistic framework can also be used to devise new AT methods and improve model robustness.
>
> ---
> Thanks for reviewing our paper and we are willing to address your further concerns before the discussion stage closes.

---

> ### Author Response · Authors · 2021-11-28
> **Please give your further opinions on our paper**
>
> Dear Reviewer MiyJ,
>
> We have updated our manuscript and replied to your comments. Would you please check whether our efforts are satisfactory and raise your score? Many thanks!
>
> Authors

---

### Official Review · Reviewer_Pwz6 · 2021-11-05

**Correctness:** 4
**Technical Novelty And Significance:** 4
**Empirical Novelty And Significance:** 3
**Recommendation:** 8
**Confidence:** 3

**Main Review:**

The paper is technically sound and brings a novel perspective to adversarial training. The paper is the first to propose sampling from an unsupervised adversarially trained model. The technical aspects of the paper are overall very impressive.

The experiment results support their claims well. The reported FID and IS of their method is also pretty good. The qualitative sample quality in Figure 1 and appendix seems worse than the JEM paper although they're getting a higher FID.

There are a few minor typos:
In related works, the second line: while standard classifiers cannot?
In section 3, it is the first time the abbreviation EBM is appearing.
In page 8 there is a Table ?? in the paragraph before Comparison with other generative models



**Summary Of The Paper:**

This paper justifies why models trained with adversarial training are good generative models by proposing a probabilistic framework. The paper is heavily influenced by the JEM paper and proposes a generalized form of analysis in that paper to include the unsupervised scenario. The generalization defines a joint distribution $p(x, z)$ over data $x$ and representations $z$. For the supervised case, each class is represented by a "center" $w_y$ and for the unsupervised one, $p(x, z) = p(x) p(z|x)$ and $p(z|x)$ is reparametrized with augmentation.

Then the authors use this probabilistic framework to analyze the adversarial training. Specifically, they show PGD is a biased form of the Langevin dynamics in their formulation. Then they show how the maximum likelihood training of their probabilistic framework is similar to adversarial training. As a result, they conclude adversarial training leads to a model with high generative power. Finally, they propose refined sampling strategies from adversarially trained models.

In the last part of the paper, they use their framework to propose an unsupervised adversarial training method followed by sampling algorithms from the produced models.

**Summary Of The Review:**

Besides the few minor typos, I think the paper meets the quality bar for ICLR. The technical analysis has some valuable novel aspects that justify well why adversarially trained models are good generative models.

---

> ### Author Response · Authors · 2021-11-18
> **Response to Reviewer Pwz6**
>
> We thank Reviewer Pwz6 for appreciating the novelty and soundness of our work. We have fixed the typos following your suggestions. Below we will address your main concerns.
>
> ---
> **Q1.** The qualitative sample quality in Figure 1 and appendix seems worse than the JEM paper although they're getting a higher FID.
>
> **A1.** Indeed there are some differences between JEM and CEM (w/ AT) samples. An eyeball comparison suggests that CEM samples have better high-level semantics than JEM, such as the clarity and consistency of the main objects, while CEM samples tend to have more low-level artifacts than JEM, such as some abnormal colors and texture.
>
> ---
> Thanks for your encouraging review and we are willing to take your additional questions.

---

### Public Comment · ~Hankook_Lee1 · 2021-11-12
**Some mathematical questions**

Dear authors,

I am very interested in this topic, so I enjoyed reading this paper. I also believe a connection exists between adversarially robust and energy-based models.

I have a few questions about this paper's mathematical derivations since some are hard to follow. Could you provide more detailed explanations for the following questions?

1. In Equations (18), (25), and (27), is the approximation $p_\theta(x)\approx p_d(x)$ reasonable (in generative modeling)? I'm wondering when such approximation makes sense or not.
2. Why Equation (12) is the same as the negative gradient of the AT loss? In Eq. (12), $x$ and $\hat{x}$ are independently drawn from $p_d$ and $p_\theta$, respectively. However, AT generates $\hat{x}$ for a given $x$. I have the same question on Equation (22).
3. Could you explain the second approximation of Equations (18), (22),  (27), and (28) in detail?

I look forward to hearing from you. Thanks!

Best, \
Hankook Lee

---

> ### Author Response · Authors · 2021-11-18
> **Response to Hankook Lee**
>
> Dear Hankook,
>
> Thanks for your interest in our work. We address your questions as follows.
>
> ---
> **Q1.** In Equations (18), is the approximation $p_{\theta}(x)\approx p_{d}(x)$ reasonable (in generative modeling)?
>
> **A1.** As we discussed in Sec 4.4, this approximation is also done in the supervised CE loss, and **it should be responsible for the lose of generative ability and adversarial robustness in standard training**. Similarly, this argument is also true for its unsupervised analogy (Eq. (18)) and **it could explain why contrastive learning models are not generative and why we need to develop unsupervised AT** (Sec 5.2). In Sec 5.1, we mainly use this approximation to draw the connection between (standard) contrastive learning and our NP-CEM, which serves as a starting point for our later development of unsupervised adversarial training and sampling methods. To wrap up, the approximation is indeed not proper for generative models, and we use it to explain why standard training  (which is equivalent to this approximation) cannot learn generative models.
>
> ---
> **Q2.** Why Equation (12) is the same as the negative gradient of the AT loss?
>
> **A1.** This is because the sampling of adversarial examples $\hat x\sim p_{\theta}(\hat x)$ can be reparameterized as two sequential steps, drawing a natural example $x\sim p_d(x)$ and adversarially perturbing it $\hat x\sim q_{\theta}(\hat x|x)$. As a result, the two terms (positive and negative) in Eq. 12 both have a $p_d(x)$, so they can be merged together (i.e., using the same $x$) without affecting the expected loss / gradient.
>
> ---
> **Q3.** Could you explain the second approximation of Equations (18), (22), (27), and (28) in detail?
>
> **A3.** We reckon that your questions on these approximations are similar to that in **Q2**. As we can merge $p_d(x)$ and $p_d(\hat x)$ in the positive and the negative phase, we can use the same set of input samples for both terms. Taking Eq. 18 for an example, we can drawn $N$ positive pairs $\{x_i,z_i\}_{i=1}^N$ from $p_d(x,z)$ (Eq. 6), and use $\{x_i\}$ as samples of $p_d(\hat x)$ as well. Besides, we can draw $K$ independent samples from $p_d(z)$ (Eq. 6) for each positive sample $x_i$. With these positive and negative samples, we can obtain our Monte Carlo estimate of the expected gradient.
>
> ---
> Hope our answers could address your concerns!
>
>
> Best,
>
> Authors

---

> > ### Public Comment · ~Hankook_Lee1 · 2021-11-19
> > **Thanks for the response! I have some follow-up questions.**
> >
> > Dear authors,
> >
> > I deeply appreciate your efforts in this response! I read it in detail and have some additional follow-up questions.
> >
> > ---
> > **Q1.** [Modified] Thanks for the detailed answer. I think I got your point.
> >
> > ---
> > **Q2.** What is $q_\theta(\hat{x}|x)$? Is $q_\theta(\hat{x}|x)$ the Langevin dynamics starting from $x$? If so, I think Eq (12) makes sense. Thanks!
> >
> > ---
> > **Q3.** I can understand now how $x_i$, $\hat{x_i}$, and $z_{ik}$ are sampled. But my main question is about the "Monte Carlo estimate". More specifically, In Eq (29) (I think it is the detailed version of Eq (18)), I'm wondering how the following approximation can be derived.
> >
> > \begin{align*}
> > \mathbb{E}_{\hat{z}\sim p_d(\hat{z})}\left[\frac{p_\theta(\hat{z}|\hat{x})}{p_\theta(\hat{z})}\nabla_\theta f\right]\approx\sum_k\frac{\exp(f(\hat{x},z_k))\nabla_\theta f}{\sum_j\exp(f(\hat{x},z_j))}
> > \end{align*}
> >
> > If I miss something, please let me know! I look forward to hearing from you. Thanks!
> >
> > Best, \
> > Hankook Lee

---

> > > ### Public Comment · ~Hankook_Lee1 · 2021-11-19
> > > **About Q1.**
> > >
> > > Sorry. I think I got your point in A1. I modified the above comment.

---

> > > ### Author Response · Authors · 2021-11-20
> > > **Response to Hankook Lee**
> > >
> > >
> > > Dear Hankook,
> > >
> > > Thanks for appreciating our feedbacks! We will address your further questions as follows.
> > >
> > >
> > > ---
> > > **Q2.** What is $q_\theta(\hat x| x)$?
> > >
> > > **A2.** Yes, it refers to the sampling process of model example $\hat x$ starting from the natural example $x$. For canonical EBM, it could be Langevin dynamics (Eq. 9); while for AT, it refers to the inner-loop adversarial attack process (Eq. 8), which, as analyzed in Sec 4.1, can also be regarded as a (biased) sampling method.
> > >
> > > ---
> > > **Q3.** How is this approximation derived in Eq. (29)?
> > > $$
> > > \mathbb{E}_{\hat{x}\sim p_d(\hat x),\hat{z}\sim p_d(\hat{z})}\left[\frac{p_\theta(\hat{z}|\hat{x})}{p_\theta(\hat{z})}\nabla_\theta f\right]\approx\sum_k\frac{\exp(f(\hat{x},z_k))\nabla_\theta f}{\sum_j\exp(f(\hat{x},z_j))}
> > > $$
> > >
> > > **A3.** We now get your point. There are indeed some typos here and we are sorry for the confusion. In particular, to achieve the final objective (Eq. 18), we need to do importance reweighting on $\hat x$ (with $\frac{p_\theta(\hat x|\hat z)}{p_\theta(\hat x)}$) instead of $\hat z$ (originally with $\frac{p_\theta(\hat z|\hat x)}{p_\theta(\hat z)}$). During the writing we swapped $x$ and $z$ in the final expression but forgot to modify the derivation accordingly. We have now fixed these typos in the revision. Note this derivation typo does not affect our conclusions.
> > >
> > > To further elaborate on your question, we note that when we can draw $K$ samples $\hat z_j\sim p_\theta(z)$, the (new) density ratio can be approximated as
> > >
> > > $$\frac{p_\theta(\hat{x}|\hat{z})}{p_\theta(\hat{x})}=\frac{p_\theta(\hat{x}|\hat{z})}{\mathbb{E}_{p_\theta(\hat z)}p_\theta(\hat{x}|\hat{z})}\approx \frac{\exp(f(\hat x, \hat z))}{\frac{1}{K}\sum_j \exp(f(\hat x,\hat z_j))},
> > > $$
> > >
> > > where the denominator is a Monte Carlo estimate of the marginal $p_\theta(\hat z)$ ($Z(\theta)$ is cancelled), and the resulting density ratio estimation is asymptotically consistent. The $1/K$ term can be cancelled by the average in the outer Monte Carlo estimate of the gradient. That being said, when deriving the standard training version in Eq. 18, we need to apply the approximation $p_\theta(\hat z)\approx p_d(\hat z)$ also in this estimate to use the $K$ negative data samples (otherwise we will require additional model samples). Now we have also made this point explicitly in Eq. 29. As discussed in **A1**, this also shows that standard training indeed introduces more bias into the gradient estimation via the approximation.
> > >
> > > ---
> > >
> > > Thanks for your careful reading and constructive comments! Hope our explanations could address your concerns.

---

### Author Response · Authors · 2021-11-18
**A Summary of Paper Updates**

We thank all reviewers for their careful reading and constructive comments. We have revised the paper with the following updates:
- **Section 1**: highlight the sampling parts in the list of contributions.
- **Section 3**: expand the full name of EBM.
- **Sec 4.2**: add explanations of Eq. (12).
- **Section 6**: add references to adversarial robustness results.
- **(News!) Appendix A.1**: derive a new **supervised AT** regularization (CR) from CEM and show that it improves adversarial robustness.
- **(News!) Appendix A.2**: derive a new **unsupervised AT** regularization (UCR) from CEM and show that it improves adversarial robustness.
- **Appendix D**: merge the experimental setup and additional results for adversarial sampling.
- Fix minor problems brought by the reviewers, such as typos.

---

### Public Comment · ~Yihong_Luo1 · 2023-08-07
**Interesting work!**

Dear authors:

   I found this work very impressive, is there any plan to release the code for reproducing your results?

---

### Decision · Program_Chairs · 2022-01-20

**Decision:**

Accept (Poster)

**Comment:**

This paper presents a probabilistic framework that explains why models trained adversarially are robust generators. It received fairly high initial scores. The reviewers thought the work was novel and interesting. They liked that the analysis provided a way to derive a novel training method and sampling algorithms. Reviewers confirmed their support of acceptance and I think this paper is clearly above the bar. Respectfully, I’d prefer that the authors don’t ask the reviewers to “raise your score”. It is up to the reviewers to make that decision.